



# Long-term trends in aerosol properties derived from AERONET measurements

Zhenyu Zhang[1], Jing Li[1], Huizheng Che[2], Yueming Dong[1], Oleg Dubovik[3], Thomas Eck[4,5], Pawan Gupta[4], Brent Holben[4], Jhoon Kim[6], Elena Lind[4], Trailokya Saud[7], Sachchida Nand Tripathi[7], and Tong Ying[1]

[1]Laboratory for Climate and Ocean-Atmosphere Studies, Department of Atmospheric and Oceanic Sciences, School of Physics, Peking University, 100871, Beijing, China
[2]State Key Laboratory of Severe Weather & Key Laboratory of Atmospheric Chemistry, Chinese Academy of Meteorological Sciences, China Meteorological Administration, 100081, Beijing, China
[3]Laboratoire d'Optique Atmosphérique, CNRS/Université de Lille, Villeneuve- d'Ascq, 59650, France
[4]NASA Goddard Space Flight Center, Greenbelt, 20771 MD, USA
[5]Goddard Earth Sciences and Technology Center, University of Maryland Baltimore County, Baltimore, MD, USA
[6]Department of Atmospheric Science, Yonsei University, Seoul, 03722, Republic of Korea
[7]Indian Institute of Technology Kanpur, Kanpur, 208016, India

Correspondence: Jing Li (jing-li@pku.edu.cn)

**Abstract.**

Over the past two decades, remarkable changes in aerosol compositions have been observed worldwide, especially over developing countries, potentially resulting in considerable changes in aerosol properties. The Aerosol Robotic Network (AERONET) offers high precision measurements of aerosol optical parameters over about 1700 stations globally, many of which have long-

term measurements for one or more decades. Here we use AERONET Level 2.0 quality assured measurements to investigate long-term trends for aerosol optical depth (AOD) and Ångström exponent (AE) trends, and quality-controlled Level 1.5 inversion products to analyze trends of absorption aerosol optical depth (AAOD) and single scattering albedo (SSA) at stations with long-term records. We also classify the aerosol properties in these sites into 6 types, and analyze the trends of each type. Results reveal decreases in AOD over the majority of the stations, except for North India and the Arabian Peninsula, where AOD

increased. AE also decreased in Europe, eastern North America, and the Middle East, but increased over South Asia and East Asia. The decreased AE over Europe and eastern North America is likely due to decreased fine-mode anthropogenic aerosols, whereas that over the Arabian Peninsula is attributed to increased dust activities. Conversely, increased AE over North India is probably attributed to increased anthropogenic emissions and decreased dust loading. Most stations in Europe, North America, East Asia, and South Asia exhibit negative trends in AAOD, whereas Solar_Village in the Arabian Peninsula has positive

trends. SSA at most stations increases and exhibits opposite trends to AAOD, but with several stations in central Europe and North America showing decreased SSA values. Trend analysis of different aerosol types further reveals the changes of different aerosol components that are related to AOD, AE, AAOD, and SSA trends. Stronger reductions in fine-mode absorbing species than that of non-absorbing aerosols are found over Europe and East Asia, whereas in eastern North America the reductions of aerosols are dominated by non-absorbing species. Increased aerosols in Kanpur over North India should be mainly comprised



of scattering species, whereas those in Solar_Village over the Arabian Peninsula are mainly dust. Weak seasonality is found in the trends of all aerosol parameters analyzed in this work.

## 1  Introduction

Aerosols are pivotal in the study of climate change due to their significant effects on the climate system. Understanding the climate effects of aerosols necessitates a comprehensive recognition of their optical and microphysical properties. Variations

in aerosol loading and aerosol properties can result in disparate climate impacts, underscoring the importance of accurately comprehending these changes. For example, changes in aerosol loading can directly influence the intensity of aerosol forcing, while a rise in aerosol absorption could even shift the aerosol forcing from negative to positive (Hansen et al., 1997), remarkably altering their climate effects. To quantify the contribution of aerosols to climate variability effectively, it is thus crucial to understand and quantify the long-term change of aerosol properties.

Studies using satellite observations revealed continuous reductions in the loading of aerosols and their precursors in Europe, North America, South America, and Africa in the past several decades, but increases over South Asia and Middle East, as well as increases in 2000s and decreases in 2010s over East Asia (Krotkov et al., 2016; Mehta et al., 2016; Zhao et al., 2017; de Meij et al., 2012; Fioletov et al., 2023; Gupta et al., 2022). In situ measurements also suggested negative scattering and absorption coefficient trends in majority of the stations which mainly located in Europe and North America, and revealed positive trends of

aerosol scattering (represented by single scattering albedo, SSA) in Asia, eastern/northern Europe, and the Arctic, and negative SSA trends in central Europe and central North America (Collaud Coen et al., 2020). As satellite observations may have drifts in long-term calibration which impact aerosol monitoring and mainly provide aerosol loading products, and the spatial coverage of in situ measurements is quite limited, ground-based remote sensing networks provide a very accurate data source to analyze trends in multiple aerosol parameters worldwide. Xia (2011) examined 79 stations within the Aerosol Robotic Network

(AERONET, Holben et al., 1998) with observations no less than six years, and found decreases in aerosol optical depth (AOD) and Ångström exponent (AE) in eastern North America and Europe. Ningombam et al. (2019) analyzed long-term AOD trends over 49 AERONET sites and 4 Sky radiometer Network (SKYNET, Takamura and Nakajima, 2004) sites, and reported decline in AOD over North-South America, Europe, the Arctic, and Australia.

However, these studies based on ground-based remote sensing data mainly focused on trends in AOD and AE, while analysis

on other aerosol optical properties, such as SSA and absorption aerosol optical depth (AAOD), is still insufficient. Other studies focusing on trends of these parameters are mainly restricted to specific stations with long-term records, which is mainly because of the limited data availability of AERONET Level 2.0 data. Li et al. (2014) utilized quality-controlled AERONET Level 1.5 inversion measurements at 54 selected stations as well as Level 2.0 solar observations at 90 selected stations worldwide for the period 2000-2013 to analyze the trends of AOD, AE, SSA, and AAOD. Decreased AOD and AAOD trends, along with

increased SSA trends, were consistently observed in Japan, Europe and North America. North America exhibited positive AE trends, whereas Europe showed negative AE trends. India was reported to experience increases in AOD, AE, and SSA. The





Arabian Peninsula was noted for experiencing increased AOD and AAOD, with decreases in AE and SSA. Eastern China was characterized by a positive SSA trend and a negative AAOD trend, without significant changes in AOD or AE.

A decade later, many regions have experienced significant changes in aerosol loading and compositions. For example, recent studies have highlighted considerable reductions in aerosol loadings in East Asia as evidenced by AERONET measurements (Yu et al., 2022; Ramachandran and Rupakheti, 2022; Eom et al., 2022) and satellite observations (Ramachandran et al., 2020; Krotkov et al., 2016; Mehta et al., 2016; Zhao et al., 2017; Fioletov et al., 2023; Li, 2020; Gupta et al., 2022). Substantial reductions in anthropogenic emissions have been observed in eastern North America (Krotkov et al., 2016), potentially contributing to a decrease in AE. Central Australia has seen reported increases in dust activities (Shao et al., 2013), aligning with observed increases in AOD and decreases in AE (Yang et al., 2021), which might also lead to positive AAOD and negative SSA trends. Some potential variations in aerosol optical properties in certain regions were not captured by Li et al. (2014), partly due to limitations in the spatial and temporal coverages of surface stations at that time, and recent changes in aerosol loadings and compositions might lead to different or reversed trends. AERONET has now expanded from 400 to over 1700 stations globally with longer records. The AERONET algorithm has also been updated to Version 3 with numerous improvements (Giles et al., 2019; Sinyuk et al., 2020). These progresses underscore the need to update trend analysis of AERONET data to capture recent shifts in aerosol optical properties and reflect advancements in data quality and network coverage.

In this study, we analyze AERONET Level 2.0 AOD and AE observations at 165 stations and Level 1.5 quality-controlled AAOD and SSA measurements at 74 stations. We also made a further attempt to categorize aerosol types and analyze the trends of each type. We hope that this study can provide a more recent reference to aerosol changes globally and facilitate the assessment of aerosol climate and environmental impacts.

## 2  Data and Methods

### 2.1  AERONET Data

The AERONET is a ground-based aerosol remote sensing network, providing long-term observations of aerosol optical and microphysical properties, covering most of the continental areas around the world (Holben et al., 1998). The AERONET AOD observations are derived from direct solar radiation at several wavelength bands mainly ranging from 340 nm to 1640 nm, while other aerosol properties, including SSA and AAOD, are derived from diffuse sky radiance at four wavelengths at 440, 675, 870, and 1020 nm (Dubovik and King, 2000). The AE parameter is calculated using AOD measurements within 440–870 nm interval (Eck et al., 1999; Giles et al., 2019). There is a series of quality assurance strategies for AERONET Level 2.0 data that ensure an AOD uncertainty of 0.01 (visible)-0.02 (UV) and an SSA uncertainty of 0.03 at $AOD_{440}$ (AOD at 440 nm) $\sim$ 0.4 (Holben et al., 2006; Giles et al., 2019; Sinyuk et al., 2020). However, as Level 2.0 quality assurance for inversion products requires a coincident AOD exceeding 0.4 at 440 nm, many stations do not have long-term Level 2.0 inversion records. Therefore, considering both the data quality and data availability, we utilize the all-point Version 3 Level 2.0 direct measurements for AOD and AE, and quality-controlled Level 1.5 almucantar inversion products (see below for the quality control scheme) for





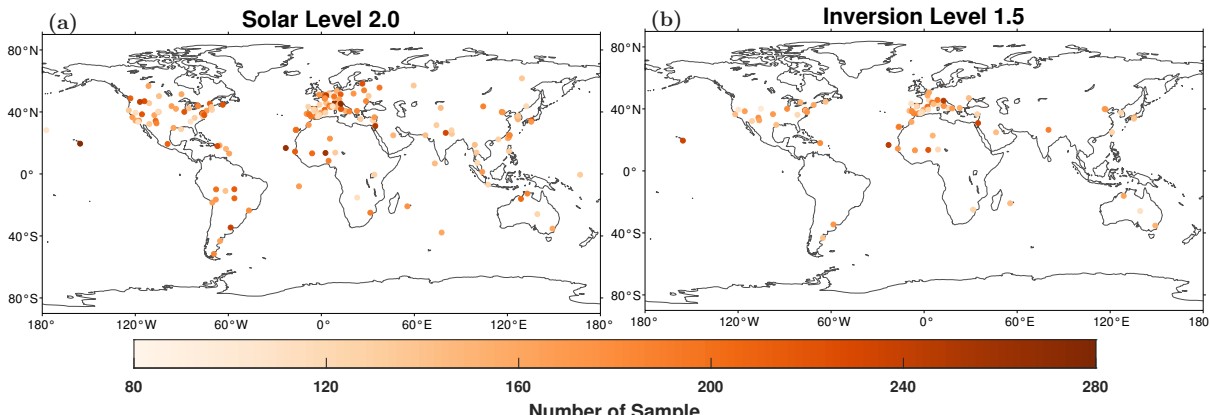

**Figure 1.** Locations of the stations selected for this study. (a) Level 2.0 solar stations, (b) Quality-controlled Level 1.5 almucantar stations. The number of monthly samples for each station is also displayed with different color.

other parameters. The uncertainty in SSA increases rapidly (exponentially) at lower AOD levels (Sinyuk et al., 2020), therefore

much of the SSA retrievals in this study have larger uncertainties of $\sim 0.03$ to $\sim 0.09$.

The stations are selected primarily based on the availability of an extensive data record for the purpose of estimating the long-term trends of aerosol properties. The Level 1.5 almucantar inversion products are first screened based on all the Level 2.0 quality assurance criteria except for the AOD threshold, such as solar zenith angle $> 50°$, sky error $< 5\%$, and coincident Level 2.0 AOD measurements. The Level 2.0 direct measurements and screened Level 1.5 almucantar inversion products are then

used to calculate monthly measurements. We calculate the median of all-point measurements to represent the monthly value only if there are more than 5 all-point measurements in at least 3 different days for that month. To ensure adequate records in trend analysis, we require the data to have at least 10 years of records and no less than 8 monthly measurements for each year during the 2000-2022 period. Specifically, the 2019-2022 data for Birdsville in Australia are eliminated for more accurate trend estimation, as these data are strongly biased due to a data filtering artifact in the quality assurance (QA) process of the

algorithm according to Giles et al. (2019), which results in a large jump in AOD (personal communication, T. Eck). This AOD artifact is caused by erroneous time stamping of the data that is greatest at some sites in Australia due to a unique data logging system utilized there. The unnatural increase in AOD for Birdsville in 2019 can be found in Yang et al. (2021). As a result, 165 stations for the direct-sun observations and 74 stations for the inversion measurements are retained for trend analysis, covering all major continents over the world. The distributions of all the selected stations as well as the number of monthly samples at

each station are presented in Fig. 1. Locations of stations mentioned in the manuscript are presented in Fig. 2.

Here we focus on analyzing AOD, SSA, and AAOD trends at 440 nm, which are noted as $AOD_{440}$, $SSA_{440}$, and $AAOD_{440}$, respectively. Trends for parameters at the other wavelengths are very similar and thus skipped. The AE is calculated from AOD measurements within the 440–870 nm wavelength range (typically including 440, 500, 675, and 870 nm), and are commonly denoted as $AE_{440\_870}$.



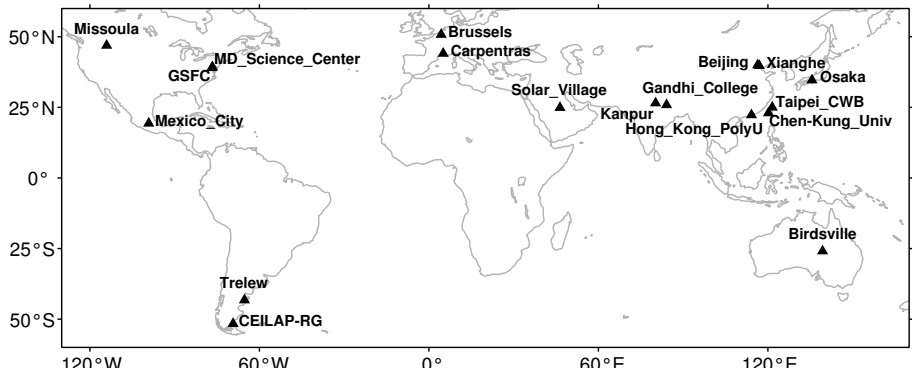

**Figure 2.** Locations of representative stations mentioned in the study.

## 2.2 Mann–Kendall test and Sen's slope for trend analysis

Here we use the Sen's slope combined with Mann-Kendall test to estimate the trend and its significance. The Mann-Kendall test (Mann, 1945; Kendall, 1975) is a nonparametric method to assess the significance of monotonic trends in a dataset without assuming any particular distribution. The slope of the trend $k$ can be estimated by the median of the set of slopes (Sen, 1968):

$$k = \text{Median}(\frac{Y_j - Y_i}{t_j - t_i}), \forall j > i \tag{1}$$

where $Y_i$ and $Y_j$ are the values of the variable at times $t_i$ and $t_j$, respectively.

The Sen's slope is a robust measurement of the trend in a dataset, and is not sensitive to outliers. As aerosol optical parameters do not follow a normal distribution, and AERONET records often have missing data, the Sen's slope is a good estimator of trends.

## 2.3 Aerosol Classification

In addition to the retrieved parameters, we also classify the observations into six aerosol types using the Fine Mode Fraction (FMF) at 550 nm and SSA at 440 nm (Lee et al., 2010). AOD and fine-mode AOD at 440, 675, 870, and 1020 nm are first interpolated to 550 nm using a second-order polynomial fit on a logarithmic scale (Eck et al., 1999). Then the $\text{FMF}_{550}$ is calculated by AOD and fine-mode AOD at 550 nm. The classification criteria for the six aerosol types (Dust, Mixture, and four fine-mode types), as well as the proportion of each type in the total number of quality-controlled Level 1.5 all-point record, are listed in Table 1. It should be noted that sea salt aerosols that typically have $\text{FMF}_{550}$ below 0.4 and $\text{SSA}_{440}$ greater than 0.95 (denoted as the Uncertain type in Table 1) are not considered in this study, because most stations are located on the mainland and sea salt aerosols only account for a negligible proportion (about 2.5%).

Each quality-controlled Level 1.5 inversion all-point measurement is classified as a specific aerosol type according to the classification criteria in Table 1. Since the records for each aerosol type are usually too few to calculate a monthly value (i.e.,



**Table 1.** Criteria of aerosol classifications defined in Lee et al. (2010).

| Aerosol type | $FMF_{550}$ | $SSA_{440}$ | Proportion |
|---|---|---|---|
| Dust | $FMF_{550} < 0.4$ | $SSA_{440} \leq 0.95$ | 14.4% |
| Mixture | $0.4 \leq FMF_{550} \leq 0.6$ | / | 17.2% |
| Non-absorbing Fine (NA) | $FMF_{550} > 0.6$ | $SSA_{440} > 0.95$ | 22.6% |
| Slightly-absorbing Fine (SA) | $FMF_{550} > 0.6$ | $0.9 < SSA_{440} \leq 0.95$ | 21.6% |
| Moderately-absorbing Fine (MA) | $FMF_{550} > 0.6$ | $0.85 < SSA_{440} \leq 0.9$ | 11.4% |
| Highly-absorbing Fine (HA) | $FMF_{550} > 0.6$ | $SSA_{440} \leq 0.85$ | 10.3% |
| Uncertain | $FMF_{550} < 0.4$ | $SSA_{440} > 0.95$ | 2.5% |

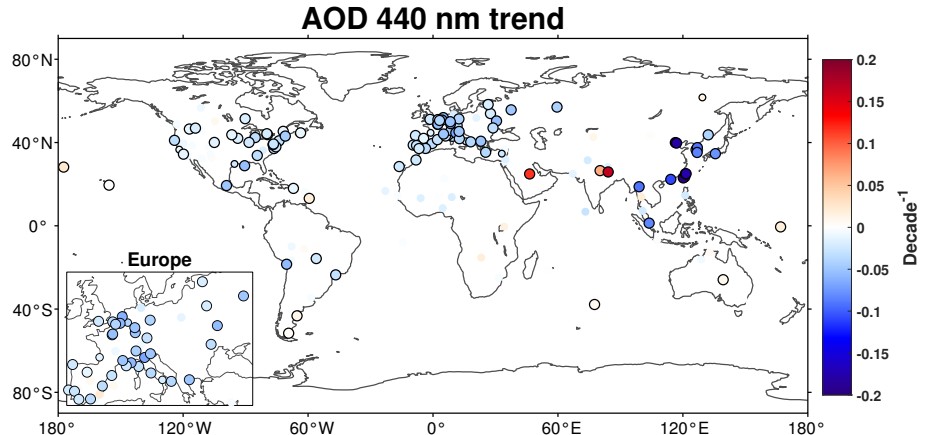

**Figure 3.** Trends of 440 nm AOD at AERONET stations. Dots without black boundary indicate trends below 90% significance level. Smaller dots with black boundary indicate trends at 90% significance, and larger dots with black boundary indicate trends at 95% significance. The magnitude of the trend has the unit of [per decade].

less than 5 measurements per month), we calculate the seasonal median values instead for the type analysis. Specifically, we use coincident Level 2.0 $AOD_{440}$ measurements to calculate the seasonal AOD and analyse its trend for each aerosol type.

## 3 Results

### 3.1 Trends for AOD and AE

The $AOD_{440}$ trends at the 165 selected AERONET stations are presented in Fig. 3. Trends surpassing the 90% significance level are marked with black circles, with larger dots denoting trends exceeding the 95% level. A global reduction of aerosol loading is found with most stations demonstrating significant negative $AOD_{440}$ trends, which is consistent with previous studies (Li et al., 2014; Xia, 2011; Ningombam et al., 2019). The $AOD_{440}$ time series at several representative sites are also shown







**Figure 4.** Time series of 440 nm AOD at several representative AERONET stations with trends at 95% significance. (a) Chen-Kung_Univ, (b) Osaka, (c) Beijing and XiangHe, (d) Solar_Village, (e) Kanpur, (f) Gandhi_College, (g) Brussels, (h) Carpentras, (i) GSFC, (j) Mexico_City.

in Fig. 4. An increased number of stations with significant trends compared to these previous studies are observed in North America, Europe, and North Africa, likely due to spatial and temporal expansion of the network in recent years. The rates of $AOD_{440}$ reduction in western Europe are not as substantial as those reported in Li et al. (2014), suggesting a decelerating trend




of aerosol reduction in Europe in recent years. This is also in line with the $AOD_{440}$ time series at representative European sites (Fig. 4g,h). Strong negative $AOD_{440}$ trends are identified at more than 10 stations in East Asia and Southeast Asia, which were previously reported as exhibiting no significant trends in global studies (Li et al., 2014; Xia, 2011; Ningombam et al., 2019). The most considerable $AOD_{440}$ reductions are observed in East China, with declines exceeding -0.1 per decade at all the five

stations (Chen-Kung_Univ, XiangHe, Taipei_CWB, Beijing, and Hong_Kong_PolyU) and almost reaching -0.2 per decade at Chen-Kung_Univ, XiangHe, and Taipei_CWB. East China was reported to have increased aerosol loading in 2000s (Yoon et al., 2012; de Meij et al., 2012; Ramachandran and Rupakheti, 2022), thus the substantial AOD reductions found in this study, which were also reported in regional studies employing recent records (Ramachandran and Rupakheti, 2022; Ramachandran et al., 2020; Yu et al., 2022; Eom et al., 2022; Li, 2020; Gupta et al., 2022), have occurred mainly in the last decade. Xianghe

and Beijing, two stations located very near to each other in East China, both possess Level 2.0 records spanning 19 years from 2000 to 2022 (Fig. 4c). However, the data record in Beijing, starting in 2001, reveals an $AOD_{440}$ trend of -0.115 per decade, whereas that in Xianghe, starting in 2004, is more recent and exhibits a larger $AOD_{440}$ decrease of -0.179 per decade, emphasizing the later years as a period of most notable $AOD_{440}$ reduction.

Significant positive $AOD_{440}$ trends are found over Kanpur and Gandhi_College in North India, Solar_Village in the Arabian

Peninsula, Birdsville in Australia, two stations in South America (Trelew and CEILAP-RG), and several oceanic island stations. Note that several of these sites (Birdsville, Trelew, CEILAP-RG, and some oceanic sites) have very low $AOD_{440}$ (typically below 0.1 for monthly values) as well as low $AOD_{440}$ variability, therefore the results in these stations are typically more uncertain. The Level 2.0 $AOD_{440}$ records at Solar_Village (Fig. 4d) ended in 2013, limiting current insights into aerosol properties in the Arabian Peninsula. Kanpur (Fig. 4e) has extensive records over the past two decades, exhibiting a positive

$AOD_{440}$ trend of 0.060 per decade. This value is close to the trends calculated from different periods in previous studies (Ramachandran and Rupakheti, 2022; Li et al., 2014; Kaskaoutis et al., 2012; Kumar et al., 2022), indicating a steady increase in $AOD_{440}$ there. Compared to previous global studies, an additional station named Gandhi_College (Fig. 4f) in northern India is observed to have a significant positive $AOD_{440}$ trend of 0.149 per decade, indicating a more pronounced increase in aerosol loading in this region. Positive AERONET $AOD_{440}$ trends over the other regions are generally weaker, with magnitudes

typically below 0.03 per decade. The positive AOD trend for Birdsville in Australia was confirmed by the independent research conducted by Yang et al. (2021), however this was a false trend resulting from a previously mentioned data screening anomaly. Hsu et al. (2012) also suggested an increase in oceanic AOD, consistent with the widespread positive trends at oceanic stations.

The $AE_{440\_870}$ parameter characterizes the wavelength dependency of AOD and closely correlates with aerosol particle size distribution. Dust particles typically have $AE_{440\_870}$ values around 0.3 or lower, and the $AE_{440\_870}$ for fine-mode particles that

are mostly anthropogenic, usually exceed 1.0 (Farahat et al., 2016; Giles et al., 2012; Russell et al., 2010; Dubovik et al., 2002). Therefore, $AE_{440\_870}$ can reflect the relative fraction of fine and coarse mode particles. The error in AE can be estimated by



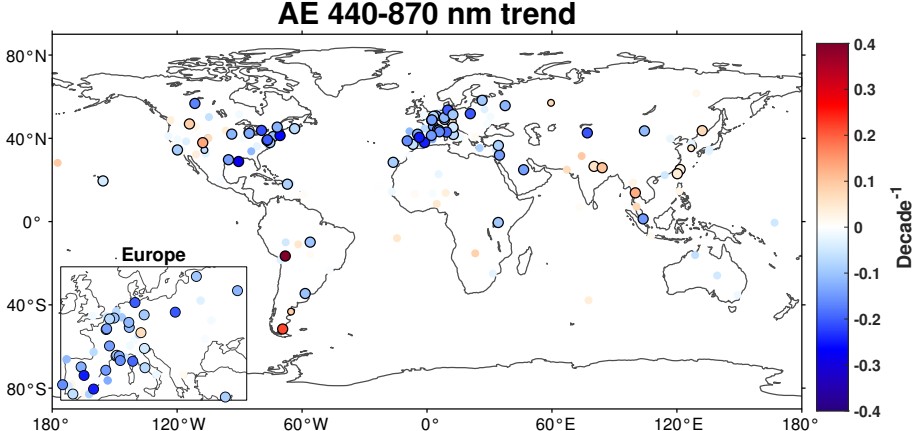

**Figure 5.** Same as Fig. 3, but with trends of AE.

the error in AOD as (Li et al., 2014; Kato et al., 2000):

$$\Delta\text{AE} = \left[ \frac{\sum\limits_{i=1}^{n} e_i^2}{(n-2)\sum\limits_{i=1}^{n}(\ln\lambda_i - \overline{\ln\lambda})^2} \right]^{\frac{1}{2}} \tag{2}$$

where $e_i$ is the error of the Ångström relationship, $n$ is the number of wavelengths $\lambda_i$ used to fit the Ångström relationship,

and $\overline{\ln\lambda}$ is the average of the logarithm of the wavelengths. $e_i$ can be estimated using the relative error of AOD ($\frac{\Delta\text{AOD}}{\text{AOD}}$), and the uncertainty of AERONET AOD ($\Delta\text{AOD}$) is considered as 0.01 here. According to Eq. 2, the uncertainty of AE is roughly inversely proportional to AOD, with larger errors at lower AOD conditions. Li et al. (2014) evaluated that the uncertainty of $\text{AE}_{440\_870}$ was 0.33 when $\text{AOD}_{440} = 0.15$, and the uncertainty would rapidly increase to 0.56 when $\text{AOD}_{440}$ decreased to 0.08. Eck et al. (1999) also demonstrated significant variability in $\text{AE}_{440\_870}$ for lower AOD, largely attributed to increased relative

errors in AOD at these low values. These results correspond to the inverse relationship between $\Delta\text{AE}$ and AOD. Therefore, it should be noted that $\text{AE}_{440\_870}$ is highly uncertain and the $\text{AE}_{440\_870}$ trends are less robust for sites with low AOD, even if the trends are statistically significant.

Significant negative $\text{AE}_{440\_870}$ trends are universally found for stations across Europe, the Mediterranean, eastern North America, the Arabian Peninsula, and Middle Asia (Fig. 5, Fig. 6). In contrast, stations in western North America, North India,

East Asia, and Southeast Asia mainly exhibit positive $\text{AE}_{440\_870}$ trends. The negative $\text{AE}_{440\_870}$ trends for Europe, the Mediterranean, and eastern North America are likely due to reductions in fine-mode anthropogenic aerosol and precursor emissions. In North India, considering the seasonal cycle of $\text{AE}_{440\_870}$ value (Fig. 6c,d), the positive $\text{AE}_{440\_870}$ trends for Kanpur and Gandhi_College primarily result from increased fine-mode anthropogenic emissions as well as decreased coarse-mode dust loading. These shifts in anthropogenic emissions have been assessed through satellite observations and emission inventories

(Pouliot et al., 2015; Szymankiewicz et al., 2021; Krotkov et al., 2016; Zhao et al., 2017; de Meij et al., 2012; Kumar et al.,





**Figure 6.** Time series of AE at several representative AERONET stations with trends at 95% significance. (a) Chen-Kung_Univ, (b) Solar_Village, (c) Kanpur, (d) Gandhi_College, (e) Brussels, (f) Carpentras, (g) GSFC, (h) Missoula.

2021), and the decline of dust loading over South Asia was also verified by satellite observations and AERONET measurements (Pandey et al., 2016, 2017; Ramachandran and Rupakheti, 2022; Kaskaoutis et al., 2011). The Arabian Peninsula is a well-known dust source (Ginoux et al., 2012) and the $AE_{440\_870}$ values are typically low (Fig. 6b), therefore the negative $AE_{440\_870}$ trend for Solar_Village is likely attributed to increased dust activities. Conversely, the increased $AE_{440\_870}$ in west-
ern North America might be partly due to both increases in biomass burning aerosols and possibly diminished dust sources. These inferences align with previous studies, as Shao et al. (2013) also reported positive dust trends in the Middle East and negative trends in North America, whereas Eck et al. (2023) and Iglesias et al. (2022) revealed increases in biomass burning emissions over western North America. Over Asia, significant positive $AE_{440\_870}$ trends are predominantly observed in the Korean Peninsula, the Indochina Peninsula, and the Taiwan Island, which might be linked to decreases in coarse-mode aerosols,





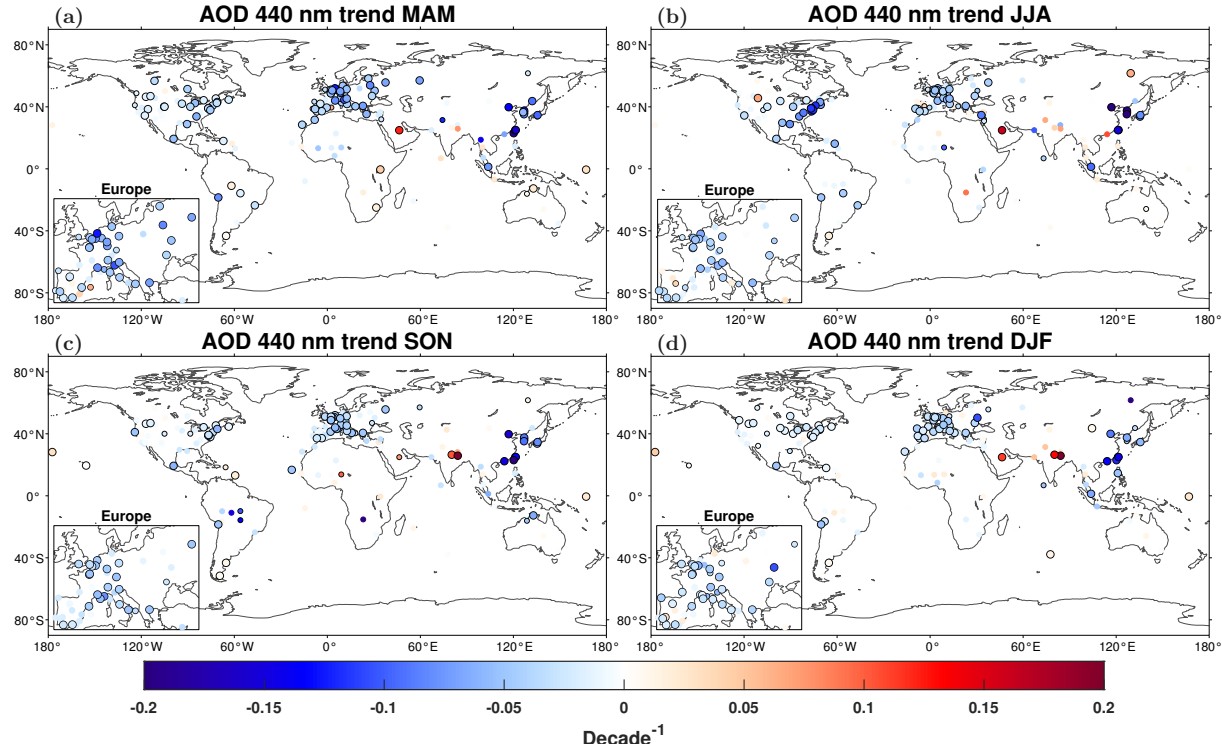

**Figure 7.** Seasonal trends of 440 nm AOD at AERONET stations. Dots without black boundary indicate trends below 90% significance level. Smaller dots with black boundary indicate trends at 90% significance, and larger dots with black boundary indicate trends at 95% significance. The magnitude of the trend has the unit of [per decade].

such as the observed decline of dust transported from the mainland (Zhang et al., 2021; Kim et al., 2017; Cho et al., 2021), and is consistent with the decline in dust emissions in the mainland reported by previous studies (Wang et al., 2021a; Wu et al., 2022; Wang et al., 2021b; Zhao et al., 2018). Notably, Beijing, Xianghe, and Hong_Kong_PolyU, despite showing substantial $AOD_{440}$ reductions, exhibit no significant $AE_{440\_870}$ trends, which might be related to reductions in both anthropogenic fine-mode aerosols and coarse-mode dust in these areas.

The spatial distribution of seasonal $AOD_{440}$ (Fig. 7) and $AE_{440\_870}$ (Fig. 8) trends is generally similar to that of annual results. Nevertheless, the magnitude of the trends could vary by season, and certain stations might exhibit significant trends only during particular seasons. This variation is largely attributed to the seasonal patterns of aerosol emissions and meteorological conditions. For example, in Europe and North America, a greater number of stations exhibit significant $AOD_{440}$ trends in MAM (Fig. 7a), while $AE_{440\_870}$ trends in DJF (Fig. 8d) are more pronounced and deviate more from the annual results. This is because aerosol concentrations are typically higher in spring and lower in winter in the Northern Hemisphere (Fig. 4), allowing for more substantial reductions in spring and more significant compositional variations in winter. However, $AE_{440\_870}$ trends at low AOD seasons, such as the more pronounced $AE_{440\_870}$ trends in winter in the Northern Hemisphere, should be treated





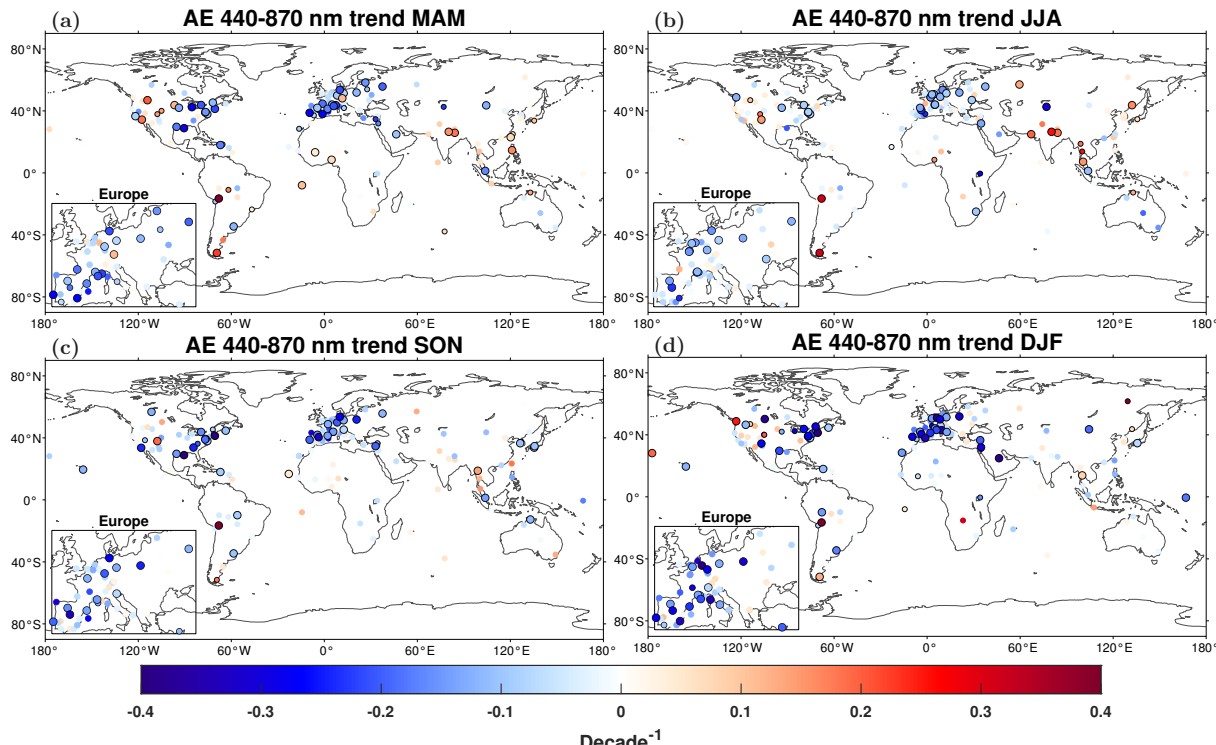

**Figure 8.** Same as Fig. 7, but with trends of AE.

with caution, because the uncertainty in $AE_{440\_870}$ becomes large at low AOD conditions. In North India, Gandhi_College and Kanpur only exhibit significant $AOD_{440}$ trends in SON (Fig. 7c) and DJF (Fig. 7d), while significant $AE_{440\_870}$ trends

predominantly occur in MAM (Fig. 8a) and JJA (Fig. 8b). We can find that $AE_{440\_870}$ values at these stations in North India significantly exceed 1.0 in SON and DJF (Fig. 6c,d), suggesting the predominance of fine-mode anthropogenic aerosols in these seasons. In contrast, $AE_{440\_870}$ values in MAM and JJA start at approximately 0.5, emphasizing the dominance of coarse-mode aerosols, and rise to about 1.0 in recent years, suggesting a largely increased fraction of fine-mode aerosols. The seasonal patterns of AOD and AE in South Asia have also been verified through multi-year observations (Adhikary et al.,

2007; Kaskaoutis et al., 2012). During the pre-monsoon (March-May) and monsoon (June-September) seasons, higher wind speeds and stronger precipitation lead to stronger dust activities and higher wet scavenging of aerosols, whereas in the post-monsoon (October-November) and winter (December-February) the meteorological conditions become reversed, with weaker dust activities and less efficient wet removal of aerosols occurred (Moorthy and Babu, 2006; Henriksson et al., 2011). As a result, in SON and DJF, the rises in anthropogenic emissions, mainly crop residue burning in post-monsoon and biofuel and

fossil fuel burning in winter (Yin, 2020; Bhardwaj et al., 2015; Venkataraman et al., 2018), have a negligible impact on changes in aerosol compositions and $AE_{440\_870}$ values, but would lead to the significant positive $AOD_{440}$ trends under less efficient wet removal. On the other hand, in MAM and JJA, stronger wet scavenging of aerosols makes the AOD trend less pronounced, and



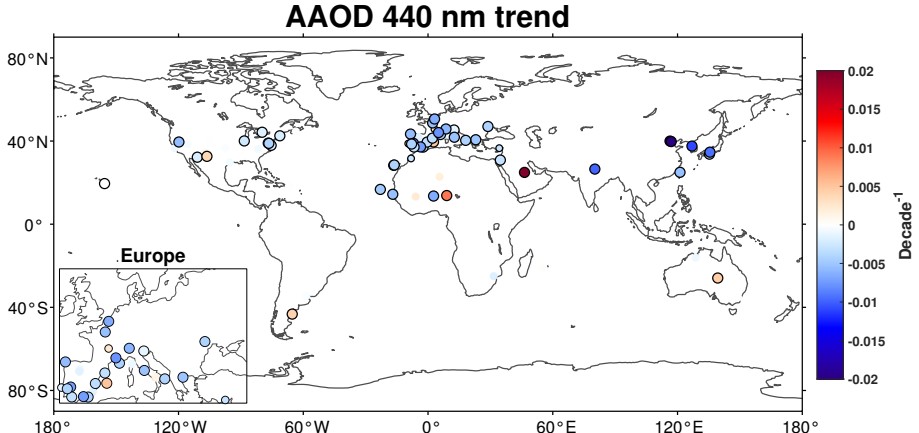

**Figure 9.** Same as Fig. 3, but with trends of AAOD.

the dominant aerosol type, dust, is mainly affected by natural variability (Kaskaoutis et al., 2012) and exhibits a negative trend (Pandey et al., 2017; Ramachandran and Rupakheti, 2022). Therefore, the increase in anthropogenic aerosols, i.e., biomass

and biofuel burning emissions, fossil fuel emissions, and industry emissions (Venkataraman et al., 2018; Ramachandran and Rupakheti, 2022), does not have a significant impact on the total $AOD_{440}$ in these two seasons, but serves to increase the fine mode fractions, leading to the insignificant $AOD_{440}$ trends and significant positive $AE_{440\_870}$ trends.

## 3.2 Trends for AAOD and SSA

AAOD and SSA both characterize the scattering and absorption properties of aerosols. AAOD represents the total aerosol
absorption optical depth, whereas SSA reflects the relative contribution of scattering to total extinction. Therefore, the AAOD trend directly reflects changes in the amount of absorbing aerosols, while the SSA trend is related to variations of both absorbing and scattering aerosols. The relationship between the two parameters can be expressed as the following equation:

$$AAOD = (1 - SSA) \times AOD \tag{3}$$

Before presenting the trends, it is important to acknowledge the uncertainties associated with AERONET inversion pa-
rameters. AERONET implements a series of quality control criteria for Level 2.0 inversion products. Under these controls, AERONET SSA have an error of $\pm 0.03$ when $AOD_{440} \sim 0.4$, and the error is even larger at lower AODs, i.e., an error of $\pm 0.05$ when $AOD_{440} \sim 0.2$, and of $\pm 0.07$ when $AOD_{440} \sim 0.1$ (Sinyuk et al., 2020). As SSA typically varies from approximately 0.8 to 1.0 (Dubovik et al., 2002; Giles et al., 2012), this error is remarkable when examining the variation of SSA and AAOD, i.e. a 0.03 error would lead to a 15% uncertainty. Therefore, the great uncertainties of these parameters should be kept
in mind when analyzing trends in this section, especially for regions with low aerosol loadings.

Similar to $AOD_{440}$, significant negative $AAOD_{440}$ trends (Fig. 9, Fig. 10) are universally found for AERONET stations in the Northern Hemisphere, especially in East Asia, North India, Europe and North America, indicating reductions in absorbing





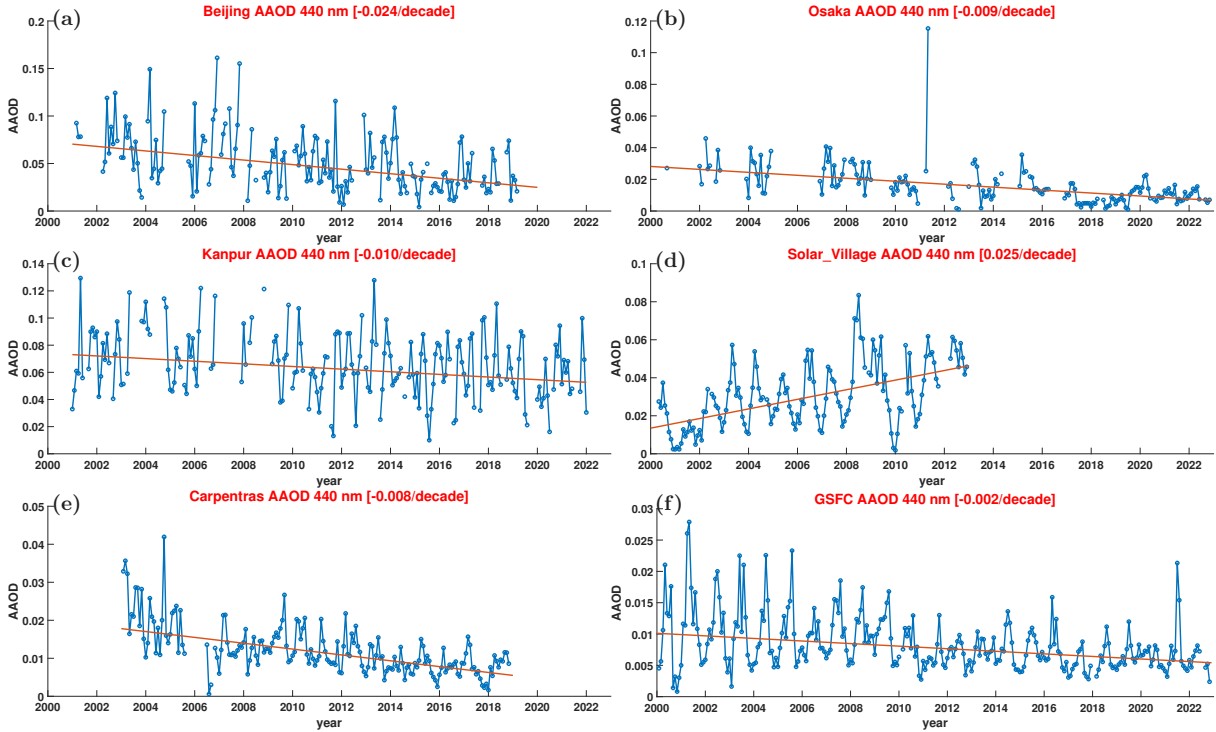

**Figure 10.** Time series of 440 nm AAOD at several representative AERONET stations with trends at 95% significance. (a) Beijing, (b) Osaka, (c) Kanpur, (d) Solar_Village, (e) Carpentras, (f) GSFC.

species, mainly primary aerosols. Conversely, significant positive $AAOD_{440}$ trend is mainly found for Solar_Village in the Arabian Peninsula (Fig. 10d), suggesting increases in absorbing aerosols. Birdsville in Australia and Trelew in southern South

America also exhibit significant positive $AAOD_{440}$ trends, but the magnitude is very low and the results are relatively uncertain because the $AAOD_{440}$ are quite small in these sites. The reductions in $AAOD_{440}$ over East Asia, Europe, North India, and North America are primarily attributed to declines in anthropogenic emissions, such as reduced black carbon (BC) and/or organic carbon (OC) emissions from fossil fuels (Ramachandran and Rupakheti, 2022; He et al., 2023; Li et al., 2024), because aerosols in these regions are mainly of the Urban/Industrial type (Li et al., 2016). Decreased dust emissions discussed in the previous

section might also be a potential contributor to the negative $AAOD_{440}$ trends in East Asia, North India, and westren North America (Shao et al., 2013; Zhang et al., 2019; Wang et al., 2021b; Ramachandran and Rupakheti, 2022; Pandey et al., 2017), but the effect might not be as substantial as that of anthropogenic emissions, since dust is not the dominant type in these regions. Significant positive $AAOD_{440}$ trend for Solar_Village in the Arabian Peninsula is likely attributed to increased dust loading. As dust mainly exhibits strong absorption for short wavelengths, AAOD trends at other channels with longer wavelengths might

not be that significant.

The $SSA_{440}$ trends (Fig. 11, Fig. 12) are generally opposite to the $AAOD_{440}$ trends, with exceptions in some stations in central Europe and North America. The majority of stations in North India, East Asia, and Europe, which also have negative

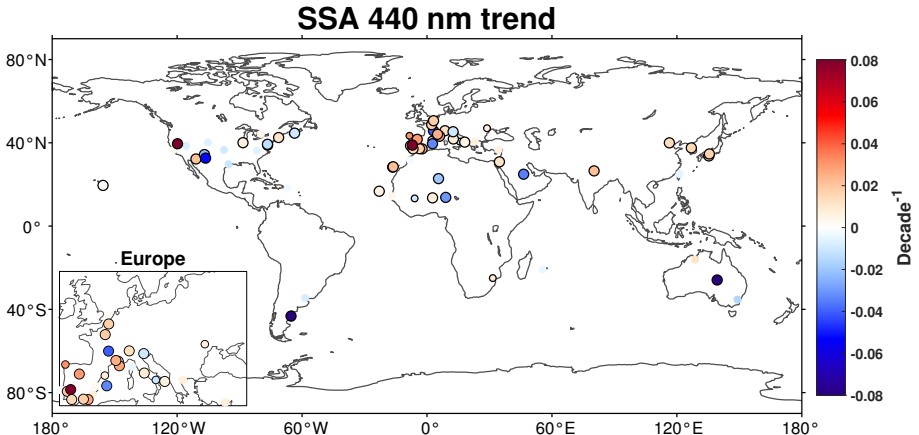

**Figure 11.** Same as Fig. 3, but with trends of SSA.

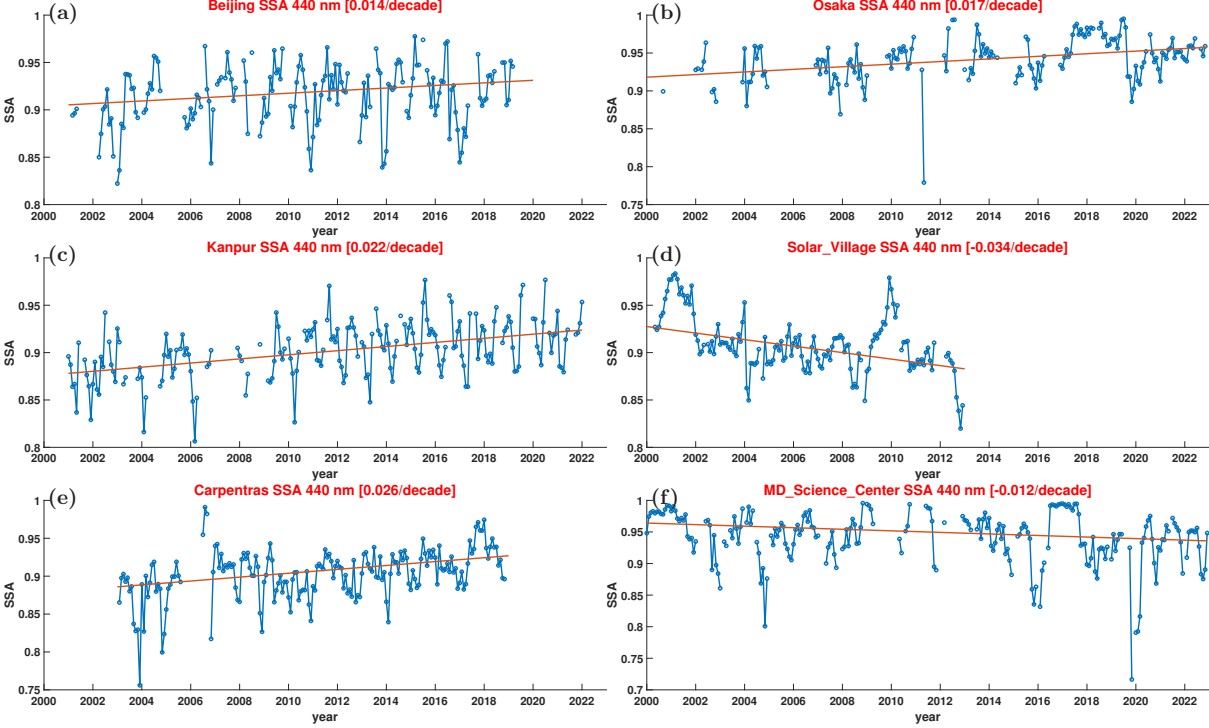

**Figure 12.** Time series of 440 nm SSA at several representative AERONET stations with trends at 95% significance. (a) Beijing, (b) Osaka, (c) Kanpur, (d) Solar_Village, (e) Carpentras, (f) MD_Science_Center.

$AAOD_{440}$ trends, exhibit significant positive $SSA_{440}$ trends, corresponding to a decrease in the fraction of absorbing aerosols over time. Remember that a rise in total aerosol loading is found for North India (Fig. 3), indicating a more pronounced in-





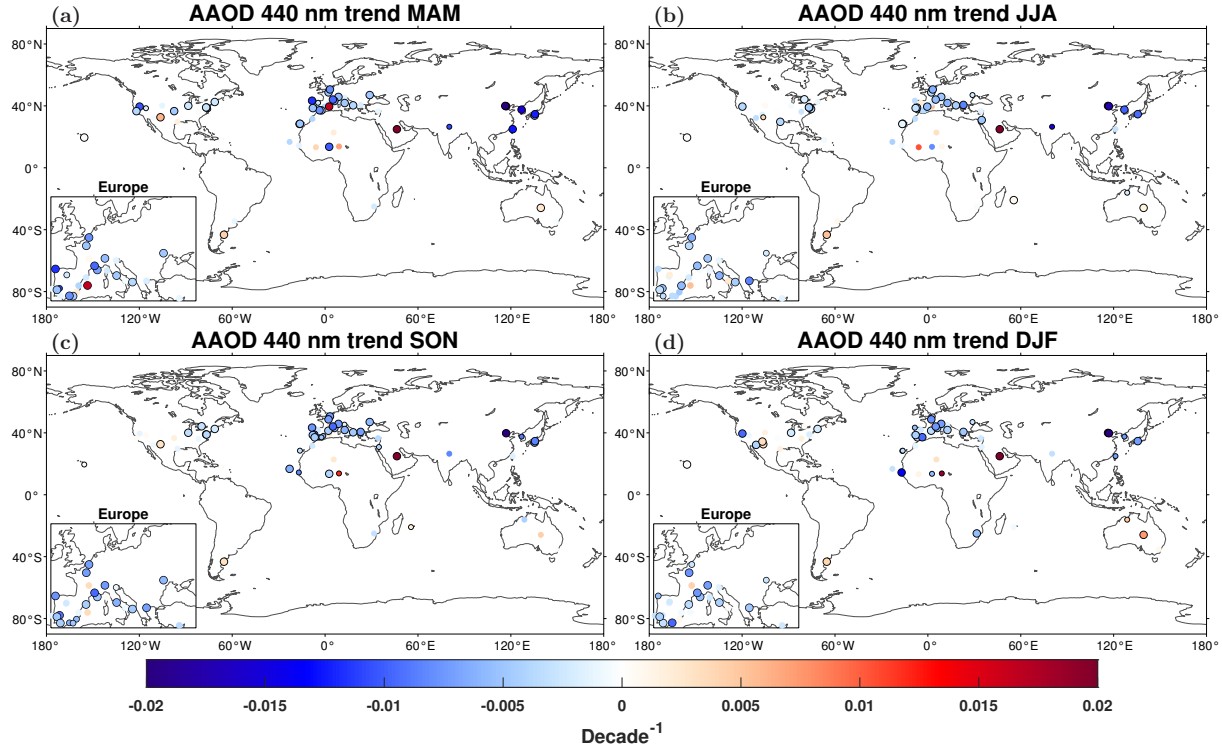

**Figure 13.** Same as Fig. 7, but with trends of AAOD.

crease in scattering aerosols, such as sulfates, than the decrease in absorbing species. For East Asia and Europe, the decreases in $AOD_{440}$ (Fig. 3) and $AAOD_{440}$ (Fig. 9) demonstrate possible reductions in both scattering and absorbing aerosols, while positive $SSA_{440}$ trends further suggest stronger reductions in asorbing species in these regions. Four stations in central Europe exhibit significant negative $SSA_{440}$ trends. $SSA_{440}$ trends show large spatial heterogeneity in North America, with five stations exhibiting significant positive trends, four with significant negative trends, and four with weak negative trends not reaching the 90% significance threshold. Stations with decreased $SSA_{440}$ over central Europe and North America exhibit only a mild decrease or even a slight increase in $AAOD_{440}$ (Fig. 9, Fig. 10f), implying that $AOD_{440}$ reduction in these regions is mainly attributed to scattering aerosols such as sulfates, thereby increasing the proportion of absorbing aerosols. This result aligns with Collaud Coen et al. (2020), which also found SSA reductions in central Europe and North America through in situ measurements, and attributed them to significant decreases in primarily scattering secondary aerosols. The substantial reductions in precursors of these scattering aerosols were also confirmed by satellite observations and emission inventories (Szymankiewicz et al., 2021; Fioletov et al., 2023; Krotkov et al., 2016; Tong et al., 2015). Positive $SSA_{440}$ trend for Solar_Village (Fig. 12d) in the Arabian Peninsula is attributed to increases in absorbing dust aerosols.

Seasonally, although some stations, mainly in North America and Europe, exhibit significant trends primarily during particular seasons, the spatial patterns of seasonal $AAOD_{440}$ (Fig. 13) and $SSA_{440}$ (Fig. 14) trends are overall similar to those





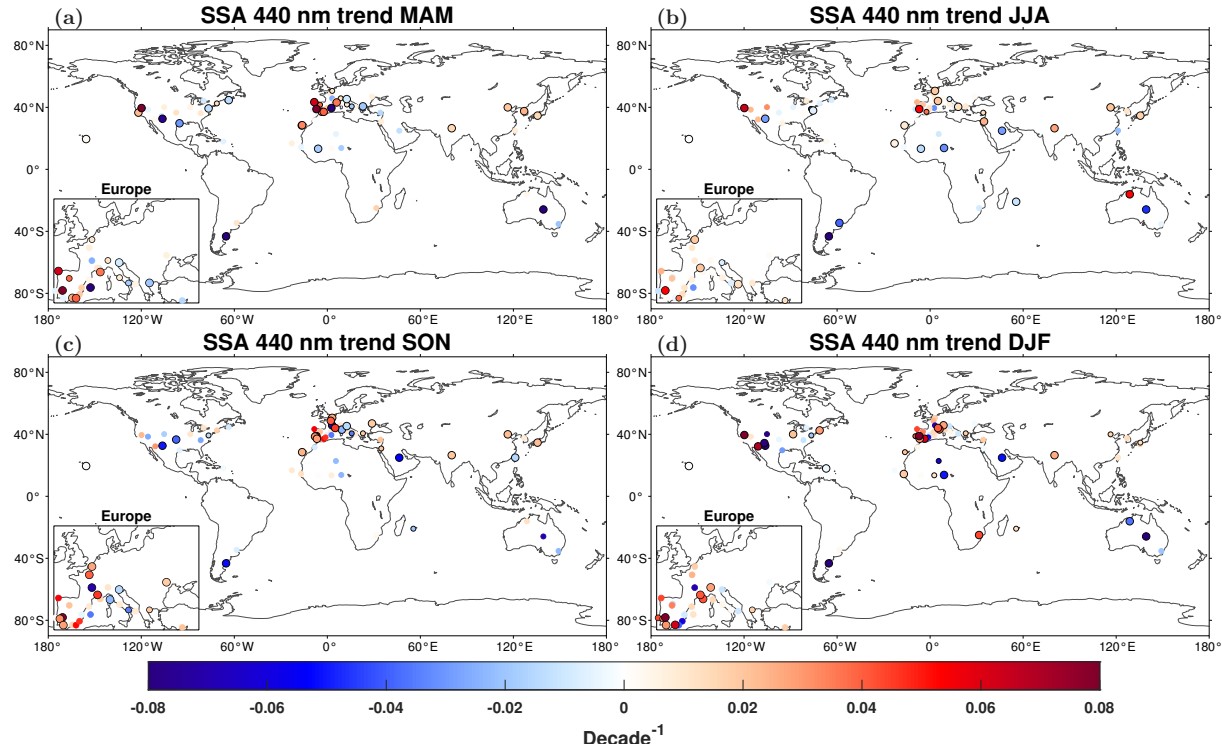

**Figure 14.** Same as Fig. 7, but with trends of SSA.

of annual trends. It is notable that Kanpur in North India exhibits stronger and more significant negative $AAOD_{440}$ trends in MAM (Fig. 13a) and JJA (Fig. 13b) where dust is the dominant aerosol type, further verifying that the decreased $AAOD_{440}$ is partly attributed to the decline in dust loading. As for $SSA_{440}$, the positive trends for Kanpur are significant at all the four seasons, indicating that the increased anthropogenic emissions in North India are mainly scattering species.

### 3.3 Aerosol type changes

To better explain the aerosol parameter changes, we make a further attempt to classify the measurements into six aerosol types as described in Sect. 2.3, and examine the long-term changes of the loadings for each type. The global $AOD_{440}$ trends of the six aerosol types are shown in Fig. 15.

Significant positive trend for dust AOD is found for Solar_Village, suggesting increased dust activities over the Arabian Peninsula, which is consistent with analysis in previous sections and other studies using satellite observations and AERONET measurements (Mehta et al., 2016; Habib et al., 2019; Sabetghadam et al., 2021; Al Otaibi et al., 2019; Li et al., 2014). We do not find significant trends over other dust sources, as dust loading can have strong decadal variability which often does not yield monotonic trends. Dust trend can also be difficult to detect when combined with fine mode anthropogenic aerosols. The Mixture type straddles the boundary between Dust type and fine-mode types and is affected by both coarse-mode and fine-





**Figure 15.** Same as Fig. 3, but with trends of AOD for 6 aerosol types. (a) dust, (b) Mixture, (c) Non-absorbing Fine, (d) Slightly-absorbing Fine, (e) Moderately-absorbing Fine, (f) Highly-absorbing Fine.

mode particles. Significant negative AOD trends in the Mixture type are mainly found over East Asia and Europe (Fig. 15b).
Since East Asia and Europe are both predominated by fine-mode aerosols (Li et al., 2016; Zhang and Li, 2019), the decreased Mixture aerosols are thus primarily due to reductions in fine-mode anthropogenic emissions.

The majority of stations in Europe, North America, and East Asia exhibit significant negative AOD trends in four fine-mode types (Fig. 15c-f), corresponding to the reduction in both absorbing and scattering anthropogenic emissions revealed by the reductions in AOD (Fig. 3) and AAOD (Fig. 9) in these regions. The great reduction in absorbing types (SA, MA and HA) is
also the possible reason for the increase of SSA (Fig. 11). It is notable that Xianghe in East Asia exhibits a significant positive





trend in non-absorbing type (NA) and even larger negative trends in the absorbing types, suggesting a great reduction in BC and/or OC emissions which might potentially lead to a shift in the predominance of aerosol type in pollution events (Zhang and Li, 2019). Eastern North America exhibits a greater reduction in non-absorbing aerosols than that in absorbing species, thus lead to an decrease in SSA (Fig. 11). Kanpur in North India exhibits significant positive trends on SA aerosols, and does not

exhibit significant trends on other types. Compared to MA and HA, the SA type is more scattering with lower BC proportion. As fine mode aerosols in Kanpur are initially absorbing types (Pandey et al., 2016), the increase in SA loading suggests a decreased proportion of BC, making the fine mode aerosols in this region more scattering.

## 4    Discussion and Conclusion

In this study, we investigate trends in aerosol optical parameters using AERONET measurements. Globally, a universal decrease

in AOD and AAOD, along with an increase in SSA, is observed at the majority of AERONET stations. The result generally aligns with the previous trend analysis using AERONET Version 2 products ending in 2013 (Li et al., 2014), highlighting the continuity of these trends over time on a global scale. Although our analysis is based on measurements at ground-based stations, coherent spatial patterns over different stations could also indicate regional features, which have also been demonstrated by satellite observations, model simulations, and emission inventories (Gupta et al., 2022; de Meij et al., 2012; Fioletov et al.,

2023; Mishchenko et al., 2007; Wei et al., 2021b, a; Yoon et al., 2016). Similar to Li et al. (2014), no significant seasonality is detected in the aerosol parameters examined in this work. Taking advantages of longer records and improved station coverage, this study identifies more detailed regional trends and finds some new spatial patterns.

Spatially, significant negative $AOD_{440}$ trends are universally observed across East Asia and Southeast Asia, which are not demonstrated by Li et al. (2014). This discrepancy indicates that the pronounced decrease in aerosols within these regions

primarily occurred over the last decade, which is also supported by satellite observations and model simulations (Fioletov et al., 2023; de Meij et al., 2012; Zhao et al., 2017; Krotkov et al., 2016; Mehta et al., 2016). The most substantial reduction in $AOD_{440}$ occurs in East China, consistent with emission inventories (Kurokawa and Ohara, 2020). Correspondingly, Li et al. (2014) also reported no significant $AE_{440\_870}$ trends, while our analysis reveals significant positive $AE_{440\_870}$ trends at coastal stations, aligning with in situ measurements (Collaud Coen et al., 2020). The increase in $AE_{440\_870}$ implies a reduction in

aerosol particle size, potentially due to decreased dust transportation from the mainland. Compared to Li et al. (2014), this study also finds that more stations in mid-latitude East Asia exhibit significant negative $AAOD_{440}$ trends and positive $SSA_{440}$ trends which are mainly attributed to decreased absorbing primary aerosols, in agreement with other independent studies utilizing AERONET data (Ramachandran and Rupakheti, 2022; Ramachandran et al., 2020; Tao et al., 2017; Yu et al., 2022; Eom et al., 2022).

Coherent significant decreases of $AOD_{440}$ and $AAOD_{440}$ found for Europe and North America are consistent with Li et al. (2014), and are in good agreement with satellite observations (Mehta et al., 2016; Zhao et al., 2017; Krotkov et al., 2016; Fioletov et al., 2023) as well as in situ measurements of recent aerosol absorbing and scattering trends (Collaud Coen et al., 2020). These trends support the ongoing efforts in emission control throughout this century. However, while Li et al. (2014)



observed smaller $AOD_{440}$ trends in North America compared to Europe, this study reports similar and weaker trends for both
regions. The decrease in anthropogenic emissions in Europe and North America started in the previous century and has led to
a significant reduction in aerosol loading (de Meij et al., 2012; Szymankiewicz et al., 2021; Rafaj et al., 2013), resulting in a
diminished rate of reduction in aerosol and aerosol precursor emissions over the last decade (Krotkov et al., 2016; Fioletov
et al., 2023; Jiang et al., 2018). Consequently, this potentially leads to smaller $AOD_{440}$ and $AAOD_{440}$ trends, alongside a
reduced discrepancy in $AOD_{440}$ trends between the two regions. Additionally, the update to AERONET Version 3 and slight
methodological differences in trend evaluation may also explain some inconsistencies in trend slope assessments. The observed
decline in $AE_{440\_870}$ and increase in $SSA_{440}$ in Europe are in line with Li et al. (2014), while the positive $AE_{440\_870}$ trends
found for the whole North America by Li et al. (2014) are mainly found in western North America in this work, with eastern
North America exhibiting negative trends. This indicates that significant reductions in dust emissions are primarily concentrated
in western North America along with increases in biomass burning emissions, with the impact on eastern areas being less
pronounced, consistent with dust monitoring results (Aryal and Evans, 2022) and trends in western North America forest fires
(Eck et al., 2023; Iglesias et al., 2022). The $SSA_{440}$ trends in this work also diverge from the coherent positive trends reported
by Li et al. (2014), because we find negative $SSA_{440}$ trends for a small proportion of European stations and for more than
half of the North American stations. This discrepancy is likely due to differences in study periods, and the trends observed in
this work align with those from in situ measurements conducted over similar periods (Collaud Coen et al., 2020), suggesting a
larger decline in scattering aerosols than absorbing species.

The positive $AOD_{440}$, $AE_{440\_870}$ and $SSA_{440}$ trends for Kanpur in North India identified by Li et al. (2014) are corroborated
in this work, suggesting increased fine-mode anthropogenic aerosol loading. Unlike the report by Li et al. (2014) which found
no significant trend in AAOD for Kanpur, our research reveals a significant negative $AAOD_{440}$ trend, indicating recent de-
creases in absorbing aerosols in the region, and we further attribute this change to both decreased anthropogenic BC emissions
and decreased dust loading according to seasonal trend analysis and type analysis, consistent with previous studies (Pandey
et al., 2016, 2017; Ramachandran and Rupakheti, 2022). The trends over North India exhibit strong seasonality, with signif-
icant positive $AOD_{440}$ trends in SON and DJF where anthropogenic aerosols are predominant, and decreased $AAOD_{440}$ and
increased $AE_{440\_870}$ in MAM and JJA where dust loading is stronger, suggesting that these seasonal trends may be associated
with the seasonal cycle of aerosol emissions and meteorological conditions. Furthermore, an additional site, Gandhi_College,
located in North India as well, also exhibits significant positive $AOD_{440}$ trends and positive $AE_{440\_870}$ trends, highlighting an
increase in fine-mode aerosols across South Asia. The $AOD_{440}$ and $AE_{440\_870}$ trends for both Kanpur and Gandhi College,
along with $AAOD_{440}$ and $SSA_{440}$ trends for Kanpur, align with independent studies utilizing AERONET measurements (Ra-
machandran and Rupakheti, 2022; Kumar et al., 2022; Kaskaoutis et al., 2012) and satellite observations (Ramachandran et al.,
2020; Kaskaoutis et al., 2011), further verifying the increment in aerosols and the alteration of aerosol compositions in North
India.

The AERONET products for Solar_Village end in 2013, therefore the trends of these aerosol optical parameters are the same
as those reported by Li et al. (2014), with positive $AOD_{440}$ and $AAOD_{440}$ trends, and negative $AE_{440\_870}$ and $SSA_{440}$ trends,



which is probably due to the increased dust activities in the Arabian Peninsula, and was also demonstrated in previous studies (Al Otaibi et al., 2019; Habib et al., 2019).

As a further step, we classify the aerosol observations into six types using $FMF_{550}$ and $SSA_{440}$, and examine the changes in aerosol loadings of each type. The trends for different aerosol types further verify the trends of AERONET parameters and offer insights into aerosol composition changes. We only find significant positive dust loading trend in the Arabian Peninsula. Significant trends mainly concentrate on fine-mode types, with declines in both absorbing types and the non-absorbing type globally, consistent with the negative $AOD_{440}$ and $AAOD_{440}$ trends. Spatially, the majority of stations in East Asia and Europe

exhibit stronger reductions in absorbing aerosols than those in non-absorbing types, whereas in eastern North America the reduction in $AOD_{440}$ is mainly attributed to non-absorbing species. The results can fully explain the changes in $SSA_{440}$, which exhibit positive trends over East Asia and Europe and negative trends over eastern North America. Significant positive SA loading trend found in North India suggests a decrease in BC proportion which leads to increased $SSA_{440}$.

This study provides insights into temporal variations in aerosol loading, optical properties, and aerosol types. Decreases in

AOD across Europe, North America, and East Asia reflect the effectiveness of emission control policies implemented in these regions. For instance, there has been a significant reduction in AOD over China in the past decade due to the Air Pollution Prevention and Control Action Plan (Gupta et al., 2022; Zhao et al., 2017). Conversely, the increase of AOD over North India and the Arabian Peninsula indicates deteriorating air quality, posing potential risks to public health. The substantial changes in SSA and AAOD observed in many regions are of concern for climate models due to their critical relationship with aerosol

climate effects, potentially influencing regional energy budget, atmospheric circulation, the water cycle, etc. Previous studies have indicated that failure to capture the increase in SSA over northern India in climate models likely contributed to their biases in simulating the negative precipitation trend in this region (Ying et al., 2023). Furthermore, trends in aerosol properties and types are crucial for satellite remote sensing applications, as many algorithms rely on assumed aerosol models clustered from AERONET observations. Updating these models to reflect changes in aerosol types may be necessary (Zhang et al., 2024).

It is important to note that our analysis extends through 2022, encompassing the COVID-19 pandemic. Previous studies have documented significant reductions in aerosol loading and notable changes in aerosol compositions due to decreased anthropogenic emissions in regions implementing lockdown policies, such as East Asia, Europe, and North America (Cao et al., 2021; Clemente et al., 2022; Liang et al., 2023; Sokhi et al., 2021). We observed abnormally low AOD values at certain stations during this period, including Xianghe and Chen-Kung_Univ (Fig. 4a, c). This could potentially lead to a negative bias

in AOD trends and contribute to discrepancies with other research on aerosol trends at these stations. However, since this period accounts for only about 10% of our total study period, and many stations lack Level 2.0 records for this time, the impact on trend analysis by COVID-19 is likely minimal at the majority of the stations.

The main purpose of this work is to update the trends in aerosol parameters with larger size of stations and longer records with respect to Li et al. (2014). We do note remarkable changes in aerosol trends over regions such as East Asia and the Southern

Hemisphere, whereas patterns in other regions remain relatively stable. Most additional stations in this study are located in Europe and North America, where the distribution of stations is already dense to deduce general features of aerosol trends in these regions. We still lack insights into aerosol trends across other regions, including Asia, Africa, South America, Australia,



and polar and oceanic regions where the spatial coverage of stations is insufficient, and some stations such as Solar_Village do not have Level 2.0 data in recent years. There is still need to establish more stations in Asia and the Southern Hemisphere to

better capture the rapid change of aerosol properties there.

*Author contributions.*  JL designed the research. TE, PG, BH, OD, and EL gathered the datasets and applied additional QAC to the data. ZZ selected the stations with long-term records, computed the trends, and analyzed the results. ZZ and JL prepared the manuscript draft. YD, TE, PG, SNT, and JK reviewed and edited the manuscript. All the other co-authors contributed to the measurements of aerosol optical properties applied in this work and to the manuscript review.

*Competing interests.*  The authors declare that they have no conflict of interest.

*Acknowledgements.*  We gratefully thank the AERONET team, especially the PIs and Co-Is and their staff of the 165 selected stations, for establishing and maintaining the sites and providing the data used in this study. The AERONET data are obtained from the AERONET website, https://aeronet.gsfc.nasa.gov/. This study is funded by the National Key Research and Development Program of China (grant no. 2023YFF0805401) and the National Natural Science Foundation of China (NSFC) Grants Nos. 42175144 and 42375121.



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
