# Peer review of "Long-term trends in aerosol properties derived from AERONET measurements"

_EGUsphere, 2024_

## Referee Comment (RC1)

**Referee comment on "Long-term trends in aerosol properties derived from AERONET measurements" by Zhang et al.**

Anonymous Referee

This article makes use of level 2 AOD AERONET data from 165 stations located worldwide to conduct a comprehensive and consistent study on trends in aerosol load (including different aerosol types) and possible changes due to the impact of emission reduction policies in developed countries, or possible increases in emissions in developing countries. This is a necessary study for the scientific community as it considerably expands the databases used in previous studies, both spatially and temporally, and includes the latest data from Version 3 of the AERONET algorithm. This information is complemented by the use of level 1.5 AERONET observations of key variables for identifying the radiative effect of aerosols, such as AAOD and SSA, taken from 74 stations located around the world. Therefore, this is a paper that provides necessary and useful information for the scientific community, is well-written, and is supported by high-quality data, perfectly aligning with the scientific objective of this journal. I thus recommend its publication in ACP with some minor and technical comments.

**Minor comments:**

General question about the stations selected in this study: I'm curious to know why polar stations are not included in the analysis, despite some of them, such as Opal or Andenes, having long-term observations. It is also evident from Figure 1 that there is a clear bias towards Europe and the United States.

Figure 2 and the stations selected for discussion: The rationale for selecting the stations displayed in this figure is not clear.

General comment about general-global trends results: Given that this paper aims to study general trends on a global scale, I wonder if it would be more appropriate to quantify the results in terms of regions. Currently, the quantification of the observed trends is done only in terms of the different stations defined (in a way that is not clear to me) in Figure 2.

A more general question: Do the authors have any ideas about the general lack of statistical significance of the results found over the African continent? While AOD and AE trends are significant over the Arabian Peninsula, suggesting a possible increase in dust activity in this region, there is no statistical significance over Africa. Some recent studies show declining DOD trends across the Sahara and the Eastern Mediterranean. Do the authors have any insights on this?

**Technical comments:**

Figure 1: Why is panel (a) labeled as "Solar" Level 2.0? I recommend using the terminology "Direct Sun," consistent with AERONET products.

Line 96: I don't understand the relevance of mentioning the "unique data logging" system used in Australia. Was there a problem with the acquisition time of the photometer?

Line 97: In line with the previous comment, the authors mention an unnatural increase in AOD in Birdsville. Are the authors referring to a diurnal cycle or to the Kciclo, as explained by Cachorro et al. (2009) and subsequent papers?

Line 120: Sea salt is not included in the aerosol typing, even though it is one of the most abundant aerosol species in Earth's atmosphere, and its hygroscopicity is an important parameter for quantifying its interaction with solar radiation.

Figure 3: I find this figure (and the following figures that use the same criterion) difficult to understand due to the exclusive use of dots. I suggest that the authors improve the figure by using different symbols to indicate varying levels of statistical significance.

Figure 4: Are the stations used in this figure selected for a specific reason? Are they chosen based on their geographical location, or do they represent significant trends? Additionally, why does Figure 4c contain two different stations in the same panel? It is difficult to distinguish between the two lines. Another suggestion is to include the country name in each subfigure label to help focus the reader's attention on the specific region discussed in the text. This suggestion could also be applied to other similar figures.

Line 135: The authors discuss the different rates of AOD reduction found in Western Europe compared to the values reported by Li et al. (2014). It would be very helpful if they could include the specific numbers found in that paper and also reference Figures 4h and 4g.

Lines 141-144: The authors state that, according to previous studies, a substantial reduction in AOD has occurred in the last decade. However, looking at Figure 4a, for instance, I see a reduction in AOD over the entire period, starting from 2002. Did the authors analyse the presence of any breakpoints in these datasets?

Lines 150 and 162: The authors mention results for "several oceanic island stations" in these two lines, while they also state that sea salt aerosols, the dominant species at these sites, are not included in the analysis. Do they expect a bias in these sites because of this omission?

Line 155: Is the AOD trend 0.066 per decade according to Figure 4e?

Line 158: Is the AOD trend 0.166 per decade according to Figure 4f?

Line 161: In line 96, the authors attribute the problems in Birdsville to the logging system, but now they attribute it to a data screening anomaly. I don't understand either of these terms. I suspect there is a calibration problem (diurnal cycle or Kciclo); can the authors confirm?

Line 194: The discussion introduced here about significant positive trends in some places in Asia is interesting. Why not include one of these stations in Figure 6?

Section 3.2: The two paragraphs starting at lines 241 and 256 are meant to provide the results related to AAOD and SSA, respectively. However, these two variables are mixed throughout both paragraphs, making it difficult for the reader to follow the discussion. I wonder if the authors could present these two pieces of information in a clearer manner.

Line 271: The authors mention a positive SSA trend in Solar Village. However, in Figure 12d, there is a negative SSA trend of -0.034 per decade. Can the authors clarify this discrepancy?

Section 3.3: I recommend using italics or quotation marks when referring to the different types of aerosols, such as "Mixture," "Dust," or "Non-absorbing," for example. I also suggest including the abbreviations SA, MA, and HA in the figure captions or somewhere in the text, since they were introduced in Table 1 (page 6).

Section 4: This section is quite long and difficult to read. Rather than focusing on highlighting the most relevant results of this study, it seems to center on the differences observed with the paper published by Li et al. (2014). I recommend summarizing and streamlining this section to emphasize the important findings of the authors.

---

## Author Comment (AC1)

**Reply to anonymous referee #1:**

We appreciate the reviewer for his/her thoughtful comments and suggestions, which are very helpful in improving our manuscript. We have carefully considered all the comments and revised the manuscript and the supplementary accordingly. Below is a point-by-point response to these comments.

**Minor comments:**

1. General question about the stations selected in this study: I'm curious to know why polar stations are not included in the analysis, despite some of them, such as Opal or Andenes, having long-term observations. It is also evident from Figure 1 that there is a clear bias towards Europe and the United States.

The stations in this study is selected based on the method described in Sect. 2.1, which requires stations to have at least 8 monthly measurements for each year for temporal representativeness. Polar stations often have no monthly measurements in winter due to inadequate sunlight and thus fail to meet the above condition. Fig. R1 shows the monthly median AOD for two polar sites, OPAL and Andenes. The two sites have no data during November-February, and the time series for OPAL is much discontinuous. Considering that the measurements at polar stations concentrate in fixed seasons (summer), we revised the standard for polar stations in the MS (lines 95-97):

*"Considering polar stations often have no monthly measurements in winter, the least number of monthly medians for each year are reduced to 4 for stations at latitudes above 65 degrees."*

Seven sites (Andenes, Barrow, Hornsund, Kangerlussuaq, PEARL, Resolute_Bay, and Thule) located in the Arctic was selected in this study, and all of them exhibit negative AOD trends, suggesting decreased AOD in the Arctic.

Indeed, the selected stations are biased towards North America and Europe. 61 and 48 stations selected in this work are located in Europe and North America respectively. This is due to the higher density and better data maintenance of the stations in Europe.

[Figure]

**Figure R1: Time series of 440 nm AOD at (a) OPAL and (b) Andenes.**

2. Figure 2 and the stations selected for discussion: The rationale for selecting the stations displayed in this figure is not clear.

Fig. 2 is only a reference for reader to know the location of stations mentioned in the MS, which has been mentioned in the MS in line 105:

*"Locations of stations mentioned in the manuscript are presented in Fig. 2."*

These stations are mentioned when analyzing particular cases (i.e., Birdsville), or showing time series as representative stations at specific regions. For the latter, the stations are selected according to their spatial representativeness, length of records, and significance and magnitude of the trends for some parameters.

Locations of other stations could be found in the supplementary, which has been mentioned in the MS in line 103:

*"Locations, trends and time series for all the stations could be found in the supplementary."*

3. General comment about general-global trends results: Given that this paper aims to study general trends on a global scale, I wonder if it would be more appropriate to quantify the results in terms of regions. Currently, the quantification of the observed trends is done only in terms of the different stations defined (in a way that is not clear to me) in Figure 2.

Thanks for the suggestion. We did attempt to calculate regional trends. However, considering the lifetime and spatial heterogeneity of aerosols, the ground-based stations have limits in spatial coverage and representativeness, and for some regions, the numbers of stations are too few to represent the entire region. Moreover, direction of regional trends could be summarized qualitatively if the trends are coherent for stations in the region, but it is difficult to quantify the magnitude and significance of the trends for a region, as trends of some stations are not significant or even opposite to those of most stations in the region. Therefore, we mainly summarized the magnitude and significance of the trends for the majority of the stations in a specific region.

As mentioned in Minor Comment #2, Fig. 2 is only a reference for the readers to know the location of stations mentioned in the MS. Only representative stations are marked. When there are very limited stations located in the region, we discuss the station-based trends. When there are many stations with coherent trends, we discuss the trend by region. We have also added a table summarizing the trends and locations of all stations in the supplementary, and mentioned it in the MS in line 103:

*"Locations, trends and time series for all the stations could be found in the supplementary."*

4. A more general question: Do the authors have any ideas about the general lack of statistical significance of the results found over the African continent? While AOD and AE trends are significant over the Arabian Peninsula, suggesting a possible increase in dust activity in this region, there is no statistical significance over Africa. Some recent studies show declining DOD trends across the Sahara and the Eastern Mediterranean. Do the authors have any insights on this?

Aerosols in West Africa are primarily composed of dust, which has strong natural variability, making it difficult to obtain a significant trend. Trends of dust loading in Sahara is still uncertain. Shao et al. (2013) reported decreased dust activities in Sahara, whereas Merdji et al. (2023) reported increased dust loading. Trends in the two studies are generally weak and not that significant. In our work, we also found the trends in aerosol parameters generally insignificant or spatially incoherent, as can be seen in the time series of stations, Banizoumbou and IER_Cinzana. AOD and AE both exhibit substantial variability, ranging from 0.2 to 1.0, and the trends are weak and insignificant.

[Figure]

**Figure R2: Left: Time series of 440 nm AOD. Right: Time series of 440-870 nm AE. (a, b) Banizoumbou, (c, d) IER_Cinzana.**

**Technical comments:**

1. Figure 1: Why is panel (a) labeled as "Solar" Level 2.0? I recommend using the terminology "Direct Sun," consistent with AERONET products.

Thanks for the suggestion. We have revised the title of Fig. 1(a) in the MS.

2. Line 96: I don't understand the relevance of mentioning the "unique data logging" system used in Australia. Was there a problem with the acquisition time of the photometer?

We are sorry for the confusion. We found a jump in AOD (more than a doubling of AOD) at Birdsville in 2019 and 2020, which coincides with the timing of the update of the algorithm. This jump could also be found in Yang et al. (2021). This is likely due to a data filtering artifact of the QA of the algorithm of Giles et al. (2019) that eliminated only the low AOD days (personal communication, T. Eck). This particular issue involves the way data are uniquely time stamped in Australia and does not occur at sites in the rest of the network.

3.   Line 97: In line with the previous comment, the authors mention an unnatural

   increase in AOD in Birdsville. Are the authors referring to a diurnal cycle or to the

   Kciclo, as explained by Cachorro et al. (2009) and subsequent papers?

The jump of AOD at Birdsville could be observed on monthly and annual time series.

According to Cachorro et al. (2008), the difference caused by KCICLO seems to be largely reduced when analyzing monthly and annual averaged data. We tend to believe that this discontinuity was caused by the algorithm upgrade. When upgrading the algorithm in the future to V4 of the AERONET database, this problem might be solved (personal communication, T. Eck).

4.   Line 120: Sea salt is not included in the aerosol typing, even though it is one of the

 most abundant aerosol species in Earth's atmosphere, and its hygroscopicity is an

 important parameter for quantifying its interaction with solar radiation.

We are sorry for the confusion. We also think that sea salt has important climate effect. In this study, sea salt is only excluded in aerosol type analysis (Sect. 3.3), because this type accounts for only 2.5% percent of total records which is too small to calculate trends, and is mainly detected at oceanic stations with low AOD levels and thus high uncertainties.

When analysing AOD, AE, AAOD, and SSA, sea salt records are not excluded. We have revised the description in lines 170-174 for clarity:

*"It should be noted that sea salt aerosols typically having $FMF_{550}$ below 0.4 and $SSA_{440}$*

*around 0.98 (included in the "Uncertain" type in Table 1) are not considered in the analysis*

*of aerosol type trends (Sect. 3.3), because most AERONET stations are located over land*

*where sea salt is not the predominant type, and sea salt aerosols only account for a*

*negligible proportion (about 2.5% for "Uncertain" type)."*

 5. Figure 3: I find this figure (and the following figures that use the same criterion) difficult to understand due to the exclusive use of dots. I suggest that the authors improve the figure by using different symbols to indicate varying levels of statistical significance.

Thanks for the suggestion. We updated the maps in the MS with different symbols to indicate differnet levels of statistical significance. Specifically, we use dots to indicate trends at 90% significance, and use triangle to represent trends below 90% significance level.

6. Figure 4: Are the stations used in this figure selected for a specific reason? Are they chosen based on their geographical location, or do they represent significant trends? Additionally, why does Figure 4c contain two different stations in the same panel? It is difficult to distinguish between the two lines. Another suggestion is to include the country name in each subfigure label to help focus the reader's attention on the specific region discussed in the text. This suggestion could also be applied to other similar figures.

Thanks for the suggestion. We have added names of the regions in each subfigure.

As mentioned in Minor Comment #2, the stations in Fig. 5 (and following similar figures) are mainly selected according to the spatial representativeness of stations, length of records, and significance and magnitude of the trends.

The two stations, Beijing and XiangHe, are combined for better comparation, as explained in the MS in lines 200-202:

*"A comparation between $AOD_{440}$ time series of XiangHe and Beijing (Fig. 5c), two stations located very close to each other in East China, would further reveal that the substantial reduction of $AOD_{440}$ mainly occurred in the later years."*

7. Line 135: The authors discuss the different rates of AOD reduction found in Western Europe compared to the values reported by Li et al. (2014). It would be very helpful if they could include the specific numbers found in that paper and also reference Figures 4h and 4g.

Thanks for the suggestion. The AOD reduction rates reported by Li et al. (2014) in Western

Europe were -0.1 per decade, while those in this work are generally -0.05 per decade. We have added these comparations in the MS in lines 185-186:

*"The rates of $AOD_{440}$ reduction in western Europe (about -0.05 per decade) are not as*

*substantial as those reported in Li et al. (2014), which was -0.1 per decade, suggesting a*

*decelerated aerosol reduction rate in Europe in recent years."*

8.   Lines 141-144: The authors state that, according to previous studies, a substantial reduction in AOD has occurred in the last decade. However, looking at Figure 4a, for instance, I see a reduction in AOD over the entire period, starting from 2002.

Did the authors analyse the presence of any breakpoints in these datasets?

We are sorry for the confusion. We have updated the result, and records before 2009 at

Chen-Kung_Univ are filtered. In fact, at most stations over East Asia, the AOD first increased or remained stable, and then decreased. The AOD reduction over these stations manly occurred after 2008 (i.e., Osaka, Beijing, and XiangHe). We have also revised the description in lines 194-198:

*"However, the trend of $AOD_{440}$ in East Asia is not coherent throughout the period of 2000-*

*2022. According to the $AOD_{440}$ time series (Fig. 5a-c), $AOD_{440}$ increased in the early*

*2000s, and decreased rapidly in the later years since around 2008, consistent with other*

*regional aerosol trend studies (Eom et al., 2022; Gupta et al., 2022; Li, 2020; Lyapustin*

*et al., 2011; Meij et al., 2012; Ramachandran et al., 2020; Ramachandran & Rupakheti,*

*2022; Yoon et al., 2012)."*

9.   Lines 150 and 162: The authors mention results for "several oceanic island stations"

in these two lines, while they also state that sea salt aerosols, the dominant species at these sites, are not included in the analysis. Do they expect a bias in these sites because of this omission?

We are sorry for the confusion. The sea salt aerosols are only excluded in aerosol type analysis in Sect. 3.3, which have been explained in the response to Minor Comment #4.

All of the AOD, AE, AAOD and SSA trend analyses in the MS include oceanic sites.

As sea salt is the dominant aerosol type at oceanic sites, the positive AOD trends for these stations could be mainly attributed to increases of sea salt aerosols. We have also added the description about increased sea salt at these oceanic sites in the MS in lines 218-220:

*"In addition to Nauru which exhibits significant positive $AOD_{440}$ trend, some other oceanic stations worldwide also exhibit positive $AOD_{440}$ trends, suggesting a widespread increase in oceanic aerosols, primarily sea salts. This result is consistent with Hsu et al. (2012) who also reported an increase in oceanic AOD."*

10. Line 155: Is the AOD trend 0.066 per decade according to Figure 4e?

Thanks for pointing this out. We are referring to the trend in Fig. 5e here, which should be 0.062 instead of 0.066. We forgot to update the value in the previous MS. We have updated Fig. 5e and revised the trend value to 0.062.

11. Line 158: Is the AOD trend 0.166 per decade according to Figure 4f?

We are sorry that this is the same issue as that in the last comment. We have updated Fig. 5f and revised the trend value to 0.167.

12. Line 161: In line 96, the authors attribute the problems in Birdsville to the logging system, but now they attribute it to a data screening anomaly. I don't understand either of these terms. I suspect there is a calibration problem (diurnal cycle or Kciclo); can the authors confirm?

We are sorry for the confusion. As detailed in Technical Comment #2, the artifact of the QA of the algorithm eliminated the low AOD records, thus likely led to a jump in AOD.

13. Line 194: The discussion introduced here about significant positive trends in some places in Asia is interesting. Why not include one of these stations in Figure 6?

Thanks for the suggestion. In fact, Chen-Kung_Univ (Fig. 6a in the previous MS draft) is one of these stations in the Taiwan Island, which exhibit significant positive AE trend in the previous MS draft. However, in the updated result, the AE trend over most of these Asia stations are not significant or coherent, therefore we revised the analysis in the MS in lines 235-238:

*"East Asia exhibits no significant $AE_{440\_870}$ trends, indicating weak changes in the ratio of fine-mode and coarse-mode aerosols. Therefore, the great decrease of aersol loading in East Asia revealed in Fig. 4 might be related to similar reductions in both anthropogenic fine-mode aerosols and coarse-mode dust in these areas."*

14. Section 3.2: The two paragraphs starting at lines 241 and 256 are meant to provide the results related to AAOD and SSA, respectively. However, these two variables are mixed throughout both paragraphs, making it difficult for the reader to follow the discussion. I wonder if the authors could present these two pieces of information in a clearer manner.

Thanks for the suggestion. We have revised the two paragraphs to separately discuss the two parameters.

15. Line 271: The authors mention a positive SSA trend in Solar Village. However, in Figure 12d, there is a negative SSA trend of -0.034 per decade. Can the authors clarify this discrepancy?

We are sorry for the confusion. The SSA trend in Solar Village is negative. We have revised the MS in lines 297-298:

*"Negative $SSA_{440}$ trend for Solar_Village (Fig. 13b) in the Arabian Peninsula is attributed to increases in absorbing dust aerosols."*

16. Section 3.3: I recommend using italics or quotation marks when referring to the different types of aerosols, such as "Mixture," "Dust," or "Non-absorbing," for example. I also suggest including the abbreviations SA, MA, and HA in the figure captions or somewhere in the text, since they were introduced in Table 1 (page 6).

Thanks for the suggestion. We have used quotation marks to refer to aerosol types in the MS.

17. Section 4: This section is quite long and difficult to read. Rather than focusing on highlighting the most relevant results of this study, it seems to center on the differences observed with the paper published by Li et al. (2014). I recommend summarizing and streamlining this section to emphasize the important findings of
the authors.

Thanks for the suggestion. We have revised Sect. 4 into a more concise expression. In
particular, we shortened the comparison with Li et al. (2014) and added more recent
references.

**References**

Cachorro, V. E., Toledano, C., Sorribas, M., Berjón, A., Frutos, A. M. de, & Laulainen, N.
(2008). An "in situ" calibration-correction procedure (KCICLO) based on AOD diurnal
cycle: Comparative results between AERONET and reprocessed (KCICLO method) AOD-
alpha data series at el arenosillo, spain. *Journal of Geophysical Research: Atmospheres*,
*113*(D2). https://doi.org/10.1029/2007jd009001

Eom, S., Kim, J., Lee, S., Holben, B. N., Eck, T. F., Park, S.-B., & Park, S. S. (2022).
Long-term variation of aerosol optical properties associated with aerosol types over east
asia using AERONET and satellite (VIIRS, OMI) data (20122019). *Atmospheric Research*,
*280*, 106457. https://doi.org/10.1016/j.atmosres.2022.106457

Gupta, G., Venkat Ratnam, M., Madhavan, B. L., & Narayanamurthy, C. S. (2022). Long-
term trends in aerosol optical depth obtained across the globe using multi-satellite
measurements. *Atmospheric Environment*, *273*, 118953.
https://doi.org/10.1016/j.atmosenv.2022.118953

Hsu, N. C., Gautam, R., Sayer, A. M., Bettenhausen, C., Li, C., Jeong, M. J., et al. (2012).
Global and regional trends of aerosol optical depth over land and ocean using SeaWiFS
measurements from 1997 to 2010. *Atmospheric Chemistry and Physics*, *12*(17), 8037–8053.
https://doi.org/10.5194/acp-12-8037-2012

Li, J. (2020). Pollution trends in china from 2000 to 2017: A multi-sensor view from space.
*Remote Sensing*, *12*(2), 208. https://doi.org/10.3390/rs12020208

Li, J., Carlson, B. E., Dubovik, O., & Lacis, A. A. (2014). Recent trends in aerosol optical properties derived from AERONET measurements. *Atmospheric Chemistry and Physics*,

*14*(22), 12271–12289. https://doi.org/10.5194/acp-14-12271-2014

Lyapustin, A., Smirnov, A., Holben, B., Chin, M., Streets, D. G., Lu, Z., et al. (2011).

Reduction of aerosol absorption in beijing since 2007 from MODIS and AERONET.

*Geophysical Research Letters*, *38*(10), L10803. https://doi.org/10.1029/2011gl047306

Meij, A. de, Pozzer, A., & Lelieveld, J. (2012). Trend analysis in aerosol optical depths and pollutant emission estimates between 2000 and 2009. *Atmospheric Environment*, *51*,

75–85. https://doi.org/10.1016/j.atmosenv.2012.01.059

Merdji, A. B., Lu, C., Xu, X., & Mhawish, A. (2023). Long-term three-dimensional distribution and transport of saharan dust: Observation from CALIPSO, MODIS, and reanalysis data. *Atmospheric Research*, *286*, 106658.

https://doi.org/10.1016/j.atmosres.2023.106658

Ramachandran, S., & Rupakheti, M. (2022). Trends in physical, optical and chemical columnar aerosol characteristics and radiative effects over south and east asia: Satellite and ground-based observations. *Gondwana Research*, *105*, 366–387.

https://doi.org/10.1016/j.gr.2021.09.016

Ramachandran, S., Rupakheti, M., & Lawrence, M. G. (2020). Aerosol-induced atmospheric heating rate decreases over south and east asia as a result of changing content and composition. *Scientific Reports*, *10*(1). https://doi.org/10.1038/s41598-020-76936-z

Shao, Y., Klose, M., & Wyrwoll, K. (2013). Recent global dust trend and connections to climate forcing. *Journal of Geophysical Research: Atmospheres*, *118*(19).

https://doi.org/10.1002/jgrd.50836

Yang, X., Zhao, C., Yang, Y., & Fan, H. (2021). Long-term multi-source data analysis about the characteristics of aerosol optical properties and types over australia. *Atmospheric*

*Chemistry and Physics*, *21*(5), 3803–3825. https://doi.org/10.5194/acp-21-3803-2021

Yoon, J., Hoyningen-Huene, W. von, Kokhanovsky, A. A., Vountas, M., & Burrows, J. P.

(2012). Trend analysis of aerosol optical thickness and ångström exponent derived from the global AERONET spectral observations. *Atmospheric Measurement Techniques*, *5*(6),

1271–1299. https://doi.org/10.5194/amt-5-1271-2012

---

## Author Comment (AC2)

**Reply to Dr. Collaud Coen**

The authors thank Dr. Collaud Coen for her detailed comments and thoughtful suggestions, which are very helpful in improving our manuscript. We have carefully considered all the comments and revised the manuscript and the supplementary accordingly. A point-by-point response to these comments is presented below.

**General comments:**

1. Methodology for trend analysis: The authors correctly chose Mann-Kendall test associated to the Sen's slope, which are both non-parametric methods. The Mann-Kendall (MK) test giving the statistically significance (ss) is however not described in the methodology section. The following points have to be clarified:

   - MK test has to be applied on serially independent data. This means that MK test without prewhitening can only be applied on time series without ss auto-correlation. In case of ss auto-correlation, prewhitening methods have to be applied. In this study, no mention of auto-correlation is found. I then require from the authors that either to report no ss auto-correlation in all the time series or to use a prewhitening method to minimize the artifacts bounded to serially dependent data. Since the authors cite Collaud Coen et al., ACP, 2020, they should also be aware of the companion paper Collaud Coen et al., AMT, 2020 (https://amt.copernicus.org/articles/13/6945/2020/ ) on MK methodology and the associated github repository (https://github.com/mannkendall) giving access to a complete MK and Sen's slope routines with prewhitening methods in R, Python and Matlab.

   - MK test also requires a homogeneous distribution, namely no seasonal cycles. The presence of seasonality in the used time series is clearly visible (e.g. Fig. 4a, b, c, d, f, Fig. 6 b, c, d, e, g and h, Fig. 10 d, e, f, Fig. 12c). Figs. 7, 8 13 and 14 clearly present the trend results for meteorological seasons. The methodology is however not described so that it is not clear if the homogeneity test between season is performed or not. The paper however describes different trends directions for
different seasons (e.g. L 200s). This has to be clarified
• Concerning seasonality, the climate specificities has also to be taken into account.
The applied seasons correspond to mid-latitude climate but not e.g. to lands with a
monsoon seasonality. The different seasonality has to be taken into account in the
analysis.
• As specified in Collaud Coen et al., AMT, 2020, the use of a lower time granularity
(e.g. daily) than month could also help to increase MK test's power
• Finally, confidence limits can also be computed and help the interpretation of the
results.

Thank you very much for the detailed comments on techniques. We revised the entire trend
analysis according to your suggestions, and the details of the method to calculate the
significance of MK test has been added in the Sect. 2.3.

The previous results did not undergo pre-whitening or seasonal homogeneity tests. We
have attempted to use the algorithm provided by Collaud Coen et al. (2020). We applied
the 3PW pre-whitening method and test the homogeneity. However, using monthly data,
the majority of stations did not pass the seasonal homogeneity test. As the main purpose of
this study is to analyse the multi-year variations of aerosol parameters, we prefer to capture
the trends on an annual scale. Therefore, we decide to calculate the annual trend using
annual mean data, which have limited auto-correlation and no seasonality. As for seasonal
results, we also calculated the seasonal means for each year, and then calculated trends for
each season using the seasonal mean time series. Estimating a valid seasonal trend also
requires at least 10 years of records. However, the updated results are less likely to show
statistical significance due to the reduced sample size.

As for the seasonal analysis, we have introduced additional season divisions for the
monsoon (South Asia) and dust source (the Arabian Peninsula and West Africa) regions.
Specifically, for South Asia, the seasons were divided into pre-monsoon (March-May),
monsoon (June-September), post-monsoon (October-November), and winter (December-
February). For the Arabian Peninsula, the seasons were divided into pre-peak (November-

February), peak (March-June), and post-peak (July-October) (Habib et al., 2019). For West
Africa, the seasons were classified as Harmattan (November-March) and summer (April-
October) (Balarabe et al., 2016; Nwofor et al., 2007). The description about season
divisions has been added in the caption of Fig. 8.

Confidence limits are also calculated, and listed in the tables in the supplementary.

2.   Homogeneity of the time series

Long-term trend analysis can only be performed on homogeneous time series. The authors
reported the case of Birdsville, where false results were reported due to false data filtering.
Which procedure was applied to check the homogeneity of the time series ? I do really
appreciate to have all time series in supplement. It's worth to have a look if we are
interested at one particular station.

Generally, I would really have a look at all time series and remove too high or too low
values (e.g. SSA below 0.6), to see if too few data are present in the first of end years so
that the time series should be shortened and if there is evident ruptures.

Here some comments on the time series:

• Ames AOD: global decrease but an increase in maxima: to check

• Amsterdam: strange high values in 201072011 and 2014

• Anmyon and Arica AOD: is there a rupture due to the long missing period ?

• Bozeman: are high AOD in 2017 and 2021 due to e.g. biomass burning ?

• CabauwAOD: I would not consider the 2 data in 2003

• Canberra: I would not take the too high 1-3 first data

• Cartel: increasing until 2006 and decreasing after a missing period in 2008-2022:
to check

• Ceilap: value > 0.15 in 2012 is doubtful

• Chen-Kung: AOD: I would not use the first two months in 2002, even if MK accept
missing data, having a full first and end years remains important. AE: idem

• Davos AOD: I would not take the 2001 Data

• Egbert: do you have an explanation for the high maxima after 2014?

• Fort-McMurry: I would not take 2005 data

•   Hamburg: AOD I would only use 2003-2016 since there is few data otherwise

•   Morin: strange AOD>4 in 2003

•   Issyk : AOD seems very high in 2021

•   Shiraham: I would stop in 2016

•   Osaka: AOD: I would not use the first two months. The maxima are in 2000-2007
are much higher than thereafter. Is there a change in 2006-2007? AAOD: the very
high data (> 0.1) should probably be invalidated and the low data end of 2017 to
mid-2019 are also strange. SSA : similar comment as for AAOD (but inverse
dependence)

•   Solar village: AOD: seems ok, SSA: the mid 2000-2002 data seems strange and too
high and max in 2010 as well as min in 2012 should be checked.

•   Gandhi college: the max at 2.5 is very strange and should be analysed. The four last
months are also much higher after a missing period. Is there a rupture in the time
series?

•   Carpentras AOD: the first ~6 months are much higher. AAOD: the 2002-2005 data
seems too high and a rupture in the time series in the missing period (2005-2006)
is probable. SSA: the three high values in 2006 should be checked

•   Mexico city: AOD: the three low data in jan-feb 2010 are strange.

•   Missoula AE: I would not use the data before 2004

•   Beijing: AAOD and SSA: I would not use the first isolated 2-3 months

•   GSFC: AAOD: the low minima in 2010 and 2011 should perhaps be investigated

•   MD Science Center SSA: the first high value until 2002 and the high values in2016-
2028 should be checked as well as the very low values in 2019.

•   Concerning the AAOD, I just looked at some station: Lille has too few data before
2007, Rome data are really too low in 2012, there is a problem, White Sands: the
increase after 2019 is so rapid that it is doubtful

•   Concerning SSA. I have the impression that SSA time series are the more uncertain.
For example, SSA at Granada is not homogeneous at all, 2012-2026 are too low.
There si various station with very low SSA (e.g. 0.5 at IMAA-Potenza, 0.3 at OHP)
that should be removed before the trend analysis. I also have the impression that

there is rupture in some dataset such as Tucson, Trelew (in 2017?), Toulon in mid-
2006, Palencia in 2007-2008.
Please have a look at all time series to improve their relevance for long-term trend analysis.

Thanks for these very detailed comments and careful observation. We have checked these
time series and changed the data filtering strategy. In the previous results using monthly
data to calculate trends, the filtering strategy were used solely to select stations with
extensive records. For the selected sites, all monthly data were used to calculate the trends,
even for years that did not meet the "at least 8 monthly measurements" criterion. Therefore,
some years may have data for only a few months, which led to discontinuity of time series,
such as Cabauw, Chen-Kung, Osaka, etc., as mentioned in the comment. In the updated
results, we switched to use the annual mean data to estimate the annual trends, and data
from the years with less than 8 monthly data have been excluded. Only those years with at
least 8 monthly data were retained to calculate annual and seasonal means. Consequently,
the updated results significantly improve the continuity and homogeneity of the data. Time
series of the following stations mentioned in the comment concerning discontinuity data
have been greatly improved: Cabauw, Chen-Kung, Davos, Fort-McMurry, Hamburg,
Shirahama, Mexico-city, Missoula, Beijing, Lille, and Rome.

We also removed outliers for the records, thus time series of the following sites having
doubtful values have been improved: Canberra, Ceilap, and Ilorin.

For SSA time series, we removed the very low values (below 0.7) from the "all-point" data.
The very low SSA values often occur alongside low AOD, meaning that these SSA values
are more uncertain.

The response to comments for some sites:

•    Ames: The AOD trend is not significant using annual mean value to calculate trend

•    Bozeman: This site is located in western North America. This region was reported to
have increased forest fires (Eck et al., 2023; Iglesias et al., 2022), thus the high AOD
in 2017 and 2021 is likely due to biomass burning.

- Cartel: This site is located in eastern North America. The AOD time series is similar to that of other stations (i.e., CCNY) in the region. Other studies using satellite observations and AERONET measurements also suggested a slight increase in AOD before 2006 (Zhao et al., 2017; Meij et al., 2012).

- Issyk: The very high AOD in 2021 is likely attributed to strong dust storms initialized by Mongolian cyclone (Yu et al., 2023).

- Osaka: The first two months have been removed because this year did not meet the "at least 8 monthly measurements" criterion (denoted as "Issue #1"). AOD increased in 2000-2007 and then reduced after 2008 in East Asia. The time series is similar to that of Li et al. (2014). The very high AAOD in 2011 has been excluded due to "Issue #1".

- Solar_village: The strange SSA values occur alongside low AOD levels in non-peak seasons when dust is less predominant, thus the SSA is influenced by anthropogenic aerosols. Moreover, low AOD levels also lead to higher SSA uncertainties. The time series is similar to that of Li et al. (2014).

- Gandhi_college: The extreme high AOD in 2011 has been removed. The four last months happen to be the winter of 2022 when AOD is high. These data have been excluded due to "Issue #1". There is no rupture in the time series using annual mean data.

- Carpentras: The AOD and AAOD time series are similar to those of Li et al. (2014). The three high SSA values in 2006 has been excluded due to "Issue #1".

Time series of other sites were also checked, and the values appear to be reasonable.

3. Results reported in a map:

The representation in a map is very useful to have an overview of the trends around the world. I have however some remarks:

- the very small trends (e.g. with AOD slopes in [-0.02, 0.02] (Fig. 3)) are in white but still sometimes ss. Since no table with all results are given, it's not easy to know if the trend are positive or negative. Moreover it means that not ss trend does not appears on the map since there is no dark circle.

• The presented results for all parameters does not correspond neither to the same
time period nor to the same length (e.g. AOD at GSFC corresponds to the 23 y trend
ending in mid-2022, whereas result from Ghandi-College correspond to 17 y trend
ending in 2021 and result from Solar in 13 y results ending in 2013) (+ Fig. 1). My
opinion is that trends with up to 10 y differences for the end point or with large
differences in the length of the time period should not be represented in a similar
way in the same figure. For example, the high positive AOD trend for Solar Village
cannot be compare with the Ghandi or Kampur trends since there is almost one
decade difference of the end time.

Thanks for the suggestion. We have edited the colobar to avoid near-white colors.
Moreover, the marks of insignificant trends have been chenged to triangles with black
boundaries. We have also added several tables in the supplementary to list the trends of
parameters of all the stations.

Time series of different sites may cover different time periods and have different length,
therefore it is hard to sort them into a few categories. Moreover, the maps in the MS are
mainly used to provide an overview of the spatial patterns about the trends. Detailed
information, such as time periods, could be observed according to time series of individual
sites included in the supplementary material.

4. Results with low AOD value and consequently larger uncertainties:

As well explained in the manuscript, low AOD values leads to high uncertainties for the
derived parameters. I think that the trends with high uncertainties should appears
differently in the map. I don't know what is the best solution. Perhaps by representing only
trends with 95% confidence level and different size as a function of the uncertainty ?

Thanks for the suggestion. It is difficult to represent the uncertainties quantitatively. The
uncertainties of AE, AAOD, and SSA are all correlated with AOD levels, but they do not
have clear relationships. Rough relationships between these parameters and AOD are
discussed in lines 127-128 of the revised manuscript:

*"According to Eq. (1), the uncertainty of AE is roughly inversely proportional to AOD,*
*with larger errors at lower AOD conditions."*

and in lines 140-141:

*AERONET SSA have an error of $\pm 0.03$ when $AOD_{440} \sim 0.4$, and the error increases*
*rapidly (exponentially) at lower AOD levels."*

All of these parameters (AE, AAOD, SSA) have higher uncertainties at lower AOD levels,
thus AOD levels could be an identifier for uncertainties qualitatively. We have added a
map of AOD in Fig. 3, and added the description about the uncertainties in lines 113-115:

*"The patterns of AOD (Fig. 3) and AOD trends (Fig. 4) should be always kept in mind*
*when analyzing trends of the other aerosol parameters, because uncertainties of the other*
*parameters are closely related to AOD level (see below), whose trend reflect changes of*
*aerosol loading."*

5.    Data used

It is not easy to understand which data are used. AERONET Solar Level 2 and AERONET
almucantar Level 1.5 data are both used, the 1.5 ones for the inversion products. L. 87-88
says that L 1.5 are similar to L 2.0 but for the AOD threshold ? meaning that no AOD
threshold are used ? It would be very helpful to have a more precise description with
eventually the mention of the level in the figures' captions.

We are sorry for the confusion. This description is generally right. AERONET Solar Level
2.0 data are used in AOD and AE analysis, whereas quality-controlled inversion Level 1.5
data are used in AAOD and SSA analysis. The quality control for Level 1.5 data that we
adopt is the same as that for Level 2.0 except the AOD threshold, as explained below.

The reason for not directly using Level 2.0 inversion data (quality assured) is the lack of
data samples (fewer than 10 stations), which is caused by the AOD threshold criterion.
This has been mentioned in the MS in lines 81-83:

*"However, as Level 2.0 quality assurance for inversion products requires a coincident*
*AOD exceeding 0.4 at 440 nm, many stations do not have enough data samples to produce*
*a long-term record."*

Nonetheless, Level 1.5 products have larger uncertainties, which is not suitable to be
directly used. As a compromise between data quality and data availability, we apply most of the Level 2.0 quality control criteria on the Level 1.5 inversion data for smaller
uncertainty, only excluding AOD threshold criterion which is an important reason for data
loss. Therefore, the amount of data samples is greatly increased.

**Minor comments:**

1.  Are all the average done with median? Are first daily medians computed and then
monthly medians or is the monthly medians computed from hourly data ?

Only monthly data is calculated with median. Annual data and seasonal data are calculated
from the monthly medians. We have added the description in the MS in lines 93-95:

*"For the years with at least 8 monthly measurements, the monthly medians are then*
*averaged to annual and seasonal means, which are used to calculate annual and seasonal*
*trends."*

The monthly medians are directly computed from AERONET all-point measurements. The
all-point data has original temporal resolution, which is calculated from every independent
observation of direct solar radiation or diffuse sky radiance.

2.  L1: there is changes in aerosol composition but also in their concentration.

Thanks for reminding. We have revised the description in line 1:

*"Over the past two decades, remarkable changes in aerosol concentrations and*
*compositions have been observed worldwide…"*

3.  L 10: I would specify that AE correspond to the wavelength dependence of AOD,
since AAOD and SSA also depend on the wavelength.

We have revised AE to *"AE (computed from the AOD within the range of 440-870 nm)"*
in line 10.

4.  L17-19: long sentence, please rephrase.

We have revised the expression in lines 18-20:

*"The reductions of aerosols in eastern North America mainly result from non-absorbing*
*species. Reductions of both fine-mode absorbing species and non-absorbing aerosols are*

*found over Europe and East Asia, but the reduction of absorbing species is stronger than*

*that of non-absorbing species."*

5. L 34: "which mainly located in …": please check the language

Thanks for reminding. We have revised the description in line 35:

*"… which are mainly located in Europe and North America"*

6. L35: It is not possible to consider SSA as representative of the scattering. Please
rephrase

Thanks for reminding. We have revised the description in lines 35-36:

*"… and revealed increased scattering aerosol fraction (represented by single scattering*
*albedo, SSA)"*

7. L84-85: Considerations on the uncertainties of the various parameters are explained
at various places in the manuscript. Please sample them at the same place so that
the reader can have a direct overview.

Thanks for the suggestion. We have regvised the MS and moved the description of these
parameters as well as their uncertainties in Sect. 2.2.

8. L 100 and Figs 1 and 2: Figs 1a and b could perhaps be merged with different color
for Level 2 and 1.5? A map (perhaps divided into continents) with all stations'name
could appears in the supplement and/or a table with the stations'coordinates.

Thanks for the suggestion. We have revised to use different colors for Level 2.0 solar and
Level 1.5 inversion measurements in Figs. 1a and 1b respectively.

We have also added several tables in the supplementaty to list the name, location, trend of
parameters of all the stations.

9. L102: does the AE corresponds to a fit including all the wavelengths between 440
and 870 nm?

The AE parameter is also a product of AERONET sun direct measurement, and is
calculated from the linear regression of AOD and wavelengths on a logarithmic scale within the range of 440-870 nm (Eck et al., 1999; Giles et al., 2019). All the AOD

measurements within the 440-870 nm are used to calculated AE (Giles et al., 2019). This has also been mentioned in the MS in lines 107-108:

*"The AE is calculated from all AOD measurements within the 440–870 nm wavelength*

*range (typically including 440, 500, 675, and 870 nm)"*

10. Eck 1999

*"Eck et al., 1999"* refers to the following research article which studied the wavelength dependence of AOD:

Eck, T. F., Holben, B. N., Reid, J. S., Dubovik, O., Smirnov, A., ONeill, N. T., et al. (1999).

Wavelength dependence of the optical depth of biomass burning, urban, and desert dust aerosols. *Journal of Geophysical Research: Atmospheres*, *104*(D24), 31333–31349.

https://doi.org/10.1029/1999jd900923

We have cited this reference in several places in the manuscript.

11. L 123: what do you mean by "all-point"?

We are sorry for the confusion. The meaning of all-point data is detailed in Minor Comment

#1. The "all-point" data is a series of AERONET products with original temporal resolution.

Detailed information could be found from the AERONET website, https://aeronet.gsfc.nasa.gov/.

12. Table 1 and L 121: Why Uncertain is not called sea salt ?

We are sorry for the confusion. We directly applied the names of the aerosol type from Lee et al. (2010), which named aerosols with $FMF_{550}$ below 0.4 and $SSA_{440}$ higher than 0.95

"Uncertain" type. The 0.95 $SSA_{440}$ threshold is mainly used to identify "Dust" aerosols, whose $SSA_{440}$ is typically 0.92-0.93 (Lee et al., 2010). Although sea salt is the coarse- mode scattering species, the $SSA_{440}$ for sea salt is typically 0.98 (Lee et al., 2010).

Therefore, the "Uncertain" type includes sea salt aerosols, but not all the "Uncertain"

aerosols are sea salt. As "Uncertain" aerosols only take a negligible proportion (2.5%), we did not further classify them into sea salt and a transitional type. We have revised the description about sea salt and "Uncertain" type in lines 170-173:

*"It should be noted that sea salt aerosols typically having $FMF_{550}$ below 0.4 and $SSA_{440}$*

*around 0.98 (included in the "Uncertain" type in Table 1) are not considered in the analysis*

*of aerosol type trends (Sect. 3.3), because most AERONET stations are located over land*

*where sea salt is not the predominant type, and sea salt aerosols only account for a*

*negligible proportion (about 2.5% for "Uncertain" type)."*

13. L125: it means that the trend results for the various aerosol types are computed from time series with three time less data points due to the seasonal median? How is the seasons defined for monsoon climate ?

We are sorry for the confusion. In the updated results, we also used annual mean AOD for each type to calculate trends. We have revised the description in lines 175-176:

*"For each aerosol type, we use coincident Level 2.0 $AOD_{440}$ measurements to calculate*

*the annual AOD and analyze its trend."*

The seasons (MAM, JJA, SON, and DJF) are defined mainly for the mid-latitude, where most AERONET stations are located. As mentioned in General Comment #1, we have re- defined seasons for monsoon and dust source regions.

14. L130-131: This is not the right causality: negative AOD trends demonstrate the global reduction of aerosol loading.

Thanks for reminding. We have revised the description in lines 181-182:

*"Significant negative $AOD_{440}$ trends are found for the majority of stations all over the*

*world, demonstrating a global reduction of aerosol loading."*

15. L135: Higher slope in Li et al. 2014 can also be due to the shortest time series leading to larger slopes due to a much lower number of data.

Thanks for reminding. We also agree that the higher slope in Li et al. (2014) might be attributed to a short data record. However, according to the time series of some European stations (Fig. 5), we could still find that the reduction of AOD has slowed down in recent years. We have revised the description in lines 185-187:

*"The rates of $AOD_{440}$ reduction in western Europe (about -0.05 per decade) are not as substantial as those reported in Li et al. (2014), which was -0.1 per decade, suggesting a decelerated aerosol reduction rate in Europe in recent years. This is also in line with the $AOD_{440}$ time series at representative European sites (Fig. 5g,h)."*

16. L139-140: In this case, it is important to know the length and end year of the time series. Do the larger slopes correspond to the shorter time series ? or to earlier end year ?

Thanks for reminding. The larger slopes indeed correspond to the shorter time series. For East Asia, Chen-Kung_Univ have only 10 years of annual records, and the AOD trend could reach -0.23 per decade. Osaka has longer AOD records, and the slope is smaller.

However, when comparing with other regions (i.e., Europe and North America), the larger slopes in East Asia do not always correspond to the shorter time series, but correlate with its higher AOD levels. For example, Beijing and XiangHe have longer records, higher AOD levels, and larger trends than Brussels and Barcelona. When reducing the same proportion of AOD, higher AOD levels would lead to larger AOD reductions, thus corresponds to larger slopes. In this case, according to the AOD time series, the most considerable AOD reductions indeed occur in East China.

17. L141-144: please rephrase

We have rephrased the description in lines 194-200:

*"However, the trend of $AOD_{440}$ in East Asia is not coherent throughout the period of 2000-2022. According to the $AOD_{440}$ time series (Fig. 5a-c), $AOD_{440}$ increased in the early 2000s, and decreased rapidly in the later years since around 2008, consistent with other regional aerosol trend studies (Eom et al., 2022; Gupta et al., 2022; Li, 2020; Lyapustin et al., 2011; Meij et al., 2012; Ramachandran et al., 2020; Ramachandran & Rupakheti, 2022; Yoon et al., 2012). This result also explains why Li et al. (2014) found no significant $AOD_{440}$ in East Asia with shorter records, as the increase of $AOD_{440}$ in the early 2000s*

*offset the reduction after 2008. When applying longer records, the continuous reduction of*
$AOD_{440}$ *after 2008 become dominant."*

18. L 147-148: does both time series have the same end year ?

AOD time series of the two sites have different end years. AOD time series of Beijing
covers the period of 2002-2018, whereas that of XiangHe covers the period of 2005-2021.
We have revised the description in the MS in lines 202-205:

*"Both statons possess Level 2.0 records spanning a period of 17 years. However, the data*
*record for Beijing, starting in 2002 and ending in 2018, reveals an $AOD_{440}$ trend of -0.175*
*per decade, whereas that for XiangHe, starting in 2005 and ending in 2021, is more recent*
*and exhibits a larger $AOD_{440}$ decrease of -0.201 per decade, emphasizing the later years*
*as a period of most notable $AOD_{440}$ reduction."*

19. L 150 and L161-162. The special case of Birdsville should be reported only once
in the paper.

Thanks for the suggestion. We have reorganized the paragraph. Discussion about Birdsville
and other sites with weak AOD trends has been moved to the second half of the paragraph
in lines 213-216:

*"Significant positive AERONET $AOD_{440}$ trends over the other regions, such as Birdsville*
*in Australia, Trelew in South America, and Nauru, an oceanic island station, are generally*
*weaker, with magnitudes typically below 0.03 per decade. As these sites have very low*
*$AOD_{440}$ (typically below 0.1 for monthly values) as well as low $AOD_{440}$ variability, the*
*results in these stations are typically more uncertain."*

20. L159-160: are all these trends ss ?

We are sorry for the confusion. We meant to indicate stations with significant positive
AOD trends here. We have clarified this in the MS in line 213:

*"Significant positive AERONET $AOD_{440}$ trends over the other regions ..."*

21. L176: which time series and seasons are less robust due to low AOD ? A map with
AOD values (or seasonal AOD) could perhaps help

We have added the map of AOD in Fig. 3, and seasonal AOD maps in the supplementary.

The description about uncertainties of analysed parameters has also been added in Sect. 2.2

in lines 113-115:

*"The patterns of AOD (Fig. 3) and AOD trends (Fig. 4) should be always kept in mind*

*when analyzing trends of the other aerosol parameters, because uncertainties of the other*

*parameters are closely related to AOD level (see below), whose trend reflect changes of*

*aerosol loading."*

22.  L179: From the map I see 2/4 stations in western North America have positive AE

trends.

We are sorry for the confusion. We have updated the result with annual mean time series (detailed in General Comment #1). In the updated map, 2 stations in western North

America have significant positive AE trends.

23.  L198-199: I have the impression that no ss AE trends is just an indicator of no modification of the size distribution. Is it right ?

Yes. The statement of reductions in both fine-mode and coarse-mode aerosols is inferred by both no ss AE trend and ss negative AOD trend. We have revised the expression in lines

235-238 for clarity:

*"East Asia exhibits no significant $AE_{440\_870}$ trends, indicating weak changes in the ratio*

*of fine-mode and coarse-mode aerosols. Therefore, the great decrease of aersol loading in*

*East Asia revealed in Fig. 4 might be related to similar reductions in both anthropogenic*

*fine-mode aerosols and coarse-mode dust in these areas."*

24.  L200-201: As mentioned in the general comments, is the homogeneity between the seasonal trends computed ?

As detailed in General comment #1, the majority of stations did not pass the seasonal homogeneity test. As the main purpose of this study is to analyse the multi-year variations of averaged aerosol parameters, we updated the results using annual mean data.

25. L204-205: Are AOD higher in spring and lower in winter for all stations in the
Northern Hemisphere? Here too a map of AOD for the various seasons could help.

Thanks for the suggestion. We have added seasonal AOD maps in the supplementary.

26. L 229: please rephrase: AAOD does not characterizes the scattering.

Thanks for reminding. We have revised the description in line 134:

*"AAOD and SSA together characterize the scattering and absorbing properties of*
*aerosols."*

27. L234-239: this should be discussed in the method/data section.

Thanks for the suggestion. The discussion about the uncertainties of AAOD and SSA have
been moved to Sect. 2.2.

28. L244: increases in either the concentration of absorbing aerosol or in the
composition (higher imaginary part of the refractive index)

Thanks for reminding. Changes in either AE, AAOD, or SSA would indicate changes in
aerosol compositions, as they suggest changes in aerosol size distribution or refractive
index or both. However, in this work, we simply regard aerosols as a mixture of absorbing
and scattering aerosols, and analyze the change of aerosol scattering and absorption
properties.

The reason for AAOD change should be analyzed together with trends of other parameters,
especially the AOD trend, which have been added in Sect. 2.2 in lines 113-115:

*"The patterns of AOD (Fig. 3) and AOD trends (Fig. 4) should be always kept in mind*
*when analyzing trends of the other aerosol parameters, because uncertainties of the other*
*parameters are closely related to AOD level (see below), whose trend reflect changes of*
*aerosol loading."*

In this case, Solar_Village have positive AOD trends, thus the increased AOD is likely
related to increases in absorbing aerosols.

29. L262: absorbing (b missing)

Thanks for reminding. We have revised it in the MS.

30. L271-272: Is there not change in BC or BrC concentrations in middle East ?

Solar_Village exhibits significant positive AOD and AAOD trends, as well as negative AE and SSA trends. This means that Solar_Village might have higher aerosol concentration, smaller FMF, and increased absorbing species. As dust is the predominant aerosol, we could infer increased dust activities according to the trends of these parameters.

Changes in BC or BrC is also possible, but we could not infer this according to the trends of AOD, AE, AAOD, and SSA, especially that the significant negative AE trend suggests decreased fine mode fraction. Aerosol type analysis in Sect. 3.3 also suggests no significant trends are found for fine-mode types. Therefore, whether BC/BrC concentration changes needs further resurach.

31. L 310: I have the impression that, e.g. SSA and AE in western North America, AOD in India or AAOD in Africa have different seasonal trends (Fig. 14).

We are sorry for the confusion. Some regions indeed have different seasonal trends for some parameters, but seasonal results are generally consistent with annual results at the majority of regions. Here we meant to express this similarity in pattern. We have revised the expression in lines 335-336 for clarity:

*"Although some regions, such as North India and western North America, have different seasonal and annual trends, the majority of regions do not exhibit significant seasonality."*

**References**

Balarabe, M., Abdullah, K., & Nawawi, M. (2016). Seasonal variations of aerosol optical properties and identification of different aerosol types based on AERONET data over sub-sahara west-africa. *Atmospheric and Climate Sciences*, *06*(01), 13–28. https://doi.org/10.4236/acs.2016.61002

Collaud Coen, M., Andrews, E., Bigi, A., Martucci, G., Romanens, G., Vogt, F. P. A., & Vuilleumier, L. (2020). Effects of the prewhitening method, the time granularity, and the time segmentation on the mann–kendall trend detection and the associated sen's slope.

*Atmospheric Measurement Techniques*, *13*(12), 6945–6964. https://doi.org/10.5194/amt-13-6945-2020

Eck, T. F., Holben, B. N., Reid, J. S., Dubovik, O., Smirnov, A., ONeill, N. T., et al. (1999). Wavelength dependence of the optical depth of biomass burning, urban, and desert dust aerosols. *Journal of Geophysical Research: Atmospheres*, *104*(D24), 31333–31349. https://doi.org/10.1029/1999jd900923

Eck, T. F., Holben, B. N., Reid, J. S., Sinyuk, A., Giles, D. M., Arola, A., et al. (2023). The extreme forest fires in california/oregon in 2020: Aerosol optical and physical properties and comparisons of aged versus fresh smoke. *Atmospheric Environment*, *305*, 119798. https://doi.org/10.1016/j.atmosenv.2023.119798

Eom, S., Kim, J., Lee, S., Holben, B. N., Eck, T. F., Park, S.-B., & Park, S. S. (2022). Long-term variation of aerosol optical properties associated with aerosol types over east asia using AERONET and satellite (VIIRS, OMI) data (20122019). *Atmospheric Research*, *280*, 106457. https://doi.org/10.1016/j.atmosres.2022.106457

Giles, D. M., Sinyuk, A., Sorokin, M. G., Schafer, J. S., Smirnov, A., Slutsker, I., et al. (2019). Advancements in the aerosol robotic network (AERONET) version 3 database automated near-real-time quality control algorithm with improved cloud screening for sun photometer aerosol optical depth (AOD) measurements. *Atmospheric Measurement Techniques*, *12*(1), 169–209. https://doi.org/10.5194/amt-12-169-2019

Gupta, G., Venkat Ratnam, M., Madhavan, B. L., & Narayanamurthy, C. S. (2022). Long-term trends in aerosol optical depth obtained across the globe using multi-satellite measurements. *Atmospheric Environment*, *273*, 118953. https://doi.org/10.1016/j.atmosenv.2022.118953

Habib, A., Chen, B., Khalid, B., Tan, S., Che, H., Mahmood, T., et al. (2019). Estimation and inter-comparison of dust aerosols based on MODIS, MISR and AERONET retrievals over asian desert regions. *Journal of Environmental Sciences*, *76*, 154–166. https://doi.org/10.1016/j.jes.2018.04.019

Iglesias, V., Balch, J. K., & Travis, W. R. (2022). U.s. Fires became larger, more frequent,
and more widespread in the 2000s. *Science Advances*, *8*(11).
https://doi.org/10.1126/sciadv.abc0020

Lee, J., Kim, J., Song, C. H., Kim, S. B., Chun, Y., Sohn, B. J., & Holben, B. N. (2010).
Characteristics of aerosol types from AERONET sunphotometer measurements.
*Atmospheric Environment*, *44*(26), 3110–3117.
https://doi.org/10.1016/j.atmosenv.2010.05.035

Li, J. (2020). Pollution trends in china from 2000 to 2017: A multi-sensor view from space.
*Remote Sensing*, *12*(2), 208. https://doi.org/10.3390/rs12020208

Li, J., Carlson, B. E., Dubovik, O., & Lacis, A. A. (2014). Recent trends in aerosol optical
properties derived from AERONET measurements. *Atmospheric Chemistry and Physics*,
*14*(22), 12271–12289. https://doi.org/10.5194/acp-14-12271-2014

Lyapustin, A., Smirnov, A., Holben, B., Chin, M., Streets, D. G., Lu, Z., et al. (2011).
Reduction of aerosol absorption in beijing since 2007 from MODIS and AERONET.
*Geophysical Research Letters*, *38*(10), L10803. https://doi.org/10.1029/2011gl047306

Meij, A. de, Pozzer, A., & Lelieveld, J. (2012). Trend analysis in aerosol optical depths
and pollutant emission estimates between 2000 and 2009. *Atmospheric Environment*, *51*,
75–85. https://doi.org/10.1016/j.atmosenv.2012.01.059

Nwofor, O. K., Chidiezie Chineke, T., & Pinker, R. T. (2007). Seasonal characteristics of
spectral aerosol optical properties at a sub-saharan site. *Atmospheric Research*, *85*(1), 38–
51. https://doi.org/10.1016/j.atmosres.2006.11.002

Ramachandran, S., & Rupakheti, M. (2022). Trends in physical, optical and chemical
columnar aerosol characteristics and radiative effects over south and east asia: Satellite and
ground-based observations. *Gondwana Research*, *105*, 366–387.
https://doi.org/10.1016/j.gr.2021.09.016

Ramachandran, S., Rupakheti, M., & Lawrence, M. G. (2020). Aerosol-induced
atmospheric heating rate decreases over south and east asia as a result of changing content
and composition. *Scientific Reports*, *10*(1). https://doi.org/10.1038/s41598-020-76936-z

Yoon, J., Hoyningen-Huene, W. von, Kokhanovsky, A. A., Vountas, M., & Burrows, J. P.
(2012). Trend analysis of aerosol optical thickness and ångström exponent derived from
the global AERONET spectral observations. *Atmospheric Measurement Techniques*, *5*(6),
1271–1299. https://doi.org/10.5194/amt-5-1271-2012

Yu, T., Xiaole, P., Yujie, J., Yuting, Z., Weijie, Y., Hang, L., et al. (2023). East asia dust
storms in spring 2021: Transport mechanisms and impacts on china. *Atmospheric Research*,
*290*, 106773. https://doi.org/10.1016/j.atmosres.2023.106773

Yu, X., Nichol, J., Lee, K. H., Li, J., & Wong, M. S. (2022). Analysis of long-term aerosol
optical properties combining AERONET sunphotometer and satellite-based observations
in hong kong. *Remote Sensing*, *14*(20), 5220. https://doi.org/10.3390/rs14205220

Zhao, B., Jiang, J. H., Gu, Y., Diner, D., Worden, J., Liou, K.-N., et al. (2017). Decadal-
scale trends in regional aerosol particle properties and their linkage to emission changes.
*Environmental Research Letters*, *12*(5), 054021. https://doi.org/10.1088/1748-
9326/aa6cb2

---

## Referee Report (RR1)

Review of « Long-term trends in aerosol properties derived from AERONET measurements" by Zhang et al.

I thank the authors for the very clear answers to my comments as well as for the changes through the manuscript. Particularly, the clear description of sources of the data,  the uncertainties in Sect. 2.2 as well as the modifications of the figures largely improves the paper. There is still some necessary improvement regarding the trend methodology:

Methodology for trend analysis:

- The authors now correctly described the MK test for the significance. They also wrote a very complete answer to my comments, but with few changes in the manuscript. First, there is no mention of the potential error due to the autocorrelation. The authors wrote in their answer that they are now using yearly medians, which is not described in Sect 2.3. In order not to use prewhitening methods, the absence of ss autorocorrelation has to be proven, since yearly data can still be autocorrelated.
- It is clear that a lot of time series do not pass the homogeneity test. The confidence level for homogeneity can then be decreases. The results for each season is however given, even if the homogeneity test is not passed. Finally, the annual trend can be easily computed since it corresponds to the median of the seasonal slopes. I still find that the use of a prewhitening method with daily or monthly data should be preferred. You could then represent the yearly trend with some different symbols is the seasonal trends are not homogeneous and give an explanation about the consequence of inhomogeneity between the seasons.
- To further weight the previous comment, Fig. 5, 7, 11 and 13 (former Fig. 4 and 6) is much less convincing and provides less information than in the previous version.
- I thank the authors to have modified the seasonal pattern, that is presently only described in the caption of Fig. 8. I think that it could be worth to add some global description in Sect 2.
- Homogeneity in the time series: I thank the authors for their answer to my comments, where they explain how they check and improve the homogeneity in the time series. A complete answer with a description on how they handle the mentioned potential problems at some stations (I mention 27 stations and they give information on 8 stations) is however missing. I do not have time to verify their work by opening the numerous files of the supplement, so that I trust them to have accomplish this fastidious work. Anyhow, it is necessary to better describe this quality control and the applied rules in Sect. 2. For exemple at line 94, the way outliers are estimated should be reported

---

## Author Response (AR2)

**Reply to Dr. Collaud Coen**

The authors thank Dr. Collaud Coen for her detailed comments and thoughtful suggestions about the methodology used in the revised manuscript. We have carefully considered the comments and revised the manuscript and the supplementary accordingly, including applying the 3PW prewhitening method to remove the autocorrelation before trend analysis, and utilizing monthly medians to calculate trends. A point-by-point response to the comments is presented below.

**Methodology for trend analysis:**

- The authors now correctly described the MK test for the significance. They also wrote a very complete answer to my comments, but with few changes in the manuscript. First, there is no mention of the potential error due to the autocorrelation. The authors wrote in their answer that they are now using yearly medians, which is not described in Sect 2.3. In order not to use prewhitening methods, the absence of ss autocorrelation has to be proven, since yearly data can still be autocorrelated.

We thank the reviewer for the suggestion. In this version, we have utilized monthly data, and applied 3PW prewhitening method (Collaud Coen et al., 2020) to remove the autocorrelation before trend analysis. We have added the description about prewhitening in the MS in Sect. 2.3:

> *It should be noted that the MK test requires serially independent data, necessitating the removal of autocorrelation from the time series before calculating trends (Collaud Coen et al., 2020; Kulkarni & Storch, 1992; Li et al., 2014; Yue et al., 2002). Several prewhitening methods are available to remove serial correlation, with Collaud Coen et al. (2020) providing a comprehensive comparison of these approaches. In this study, we apply the 3PW method developed by Collaud Coen et al. (2020) to eliminate autocorrelation before computing the trend.*

•       It is clear that a lot of time series do not pass the homogeneity test. The
    confidence level for homogeneity can then be decreases. The results for each
    season is however given, even if the homogeneity test is not passed. Finally, the
    annual trend can be easily computed since it corresponds to the median of the
    seasonal slopes. I still find that the use of a prewhitening method with daily or
    monthly data should be preferred. You could then represent the yearly trend with
    some different symbols is the seasonal trends are not homogeneous and give an
    explanation about the consequence of inhomogeneity between the seasons.

We thank the reviewer for the suggestion. We have applied the 3PW prewhitening
method with monthly data in trend analysis, and have added the related description in
Sect. 2.3:

*It should be noted that the MK test requires serially independent data,*
*necessitating the removal of autocorrelation from the time series before*
*calculating trends (Collaud Coen et al., 2020; Kulkarni & Storch, 1992; Li et al.,*
*2014; Yue et al., 2002). Several prewhitening methods are available to remove*
*serial correlation, with Collaud Coen et al. (2020) providing a comprehensive*
*comparison of these approaches. In this study, we apply the 3PW method*
*developed by Collaud Coen et al. (2020) to eliminate autocorrelation before*
*computing the trend.*

The seasonal homogeneity is also shown in the yearly trend maps (Fig. 4, 6, 10, 12), with
magenta boundaries representing trends passing homogeneity test. The discussion about
seasonal homogeneity have been added in sections of seasonal analysis.

•       To further weight the previous comment, Fig. 5, 7, 11 and 13 (former Fig. 4 and 6)
    is much less convincing and provides less information than in the previous
    version.

The time series figures (Fig. 5, 7, 11, and 13) have also been replaced by monthly time
series.

 • I thank the authors to have modified the seasonal pattern, that is presently only described in the caption of Fig. 8. I think that it could be worth to add some global description in Sect 2.

Thanks for the suggestion. We have added the description about the seasonal pattern in

Sect. 2.3:

*Aerosol parameters typically exhibit strong seasonality, which should be taken*

*into accounted in the analysis. We conduct seasonal MK tests and calculate*

*seasonal trends on the prewhitened time series, and then derive the annual trend*

*as the median of seasonal trends (Hirsch et al., 1982; Hirsch & Slack, 1984). The*

*homogeneity of seasonal trend is also tested, and the results are marked in the*

*annual trend maps. The definition of the seasons is primarily based on regional*

*climatic characteristics. Specifically, seasons for South Asia are diveded into pre-*

*monsoon (March-May), monsoon (June-September), post-monsoon (October-*

*November), and winter (December-February). For the Arabian Peninsula, the*

*seasons are categorized as pre-peak (November-February), peak (March-June),*

*and post-peak (July-October) (Habib et al., 2019). In West Africa, the seasons are*

*classified as Harmattan (November-March) and summer (April-October)*

*(Balarabe et al., 2016; Nwofor et al., 2007). For the other regions, the standard*

*seasonal divisions of spring (March-May), summer (June-August), autumn*

*(September-November), and winter (December-February) are applied.*

• Homogeneity in the time series: I thank the authors for their answer to my comments, where they explain how they check and improve the homogeneity in the time series. A complete answer with a description on how they handle the mentioned potential problems at some stations (I mention 27 stations and they give information on 8 stations) is however missing. I do not have time to verify their work by opening the numerous files of the supplement, so that I trust them to have accomplish this fastidious work. Anyhow, it is necessary to better describe this quality control and the applied rules in Sect. 2. For exemple at line 94, the way outliers are estimated should be reported.

Thank you for your suggestion! In our previous response, we systematically checked all
the stations, including the 27 stations highlighted by the reviewer. Indeed, we addressed
the identified issues for 22 stations, and listed detailed description in the previous
response. Fourteen stations (11 stations concerning discontinuity data and 3 stations
concerning outliers) were categorized and combined, and the other eight stations with
complicated problems were listed separately. Records of the other five stations appear to
be resonable. Here we copy the response to these 14 stations with common problems
mentioned in our previous response:

*Only those years with at least 8 monthly data were retained to calculate annual*
*and seasonal means. Consequently, the updated results significantly improve the*
*continuity and homogeneity of the data. Time series of the following stations*
*mentioned in the comment concerning discontinuity data have been greatly*
*improved: Cabauw, Chen-Kung, Davos, Fort-McMurry, Hamburg, Shirahama,*
*Mexico-city, Missoula, Beijing, Lille, and Rome.*

*We also removed outliers for the records, thus time series of the following sites*
*having doubtful values have been improved: Canberra, Ceilap, and Ilorin.*

We have detailed the quality control criteria in Sect. 2.1:

*Long-term trend analysis necessitates homogeneous time series, and outliers*
*would influence the result. We first check the records, removing invalid and*
*abnormally high or low values (such as SSA below 0.7 for all stations, and AOD*
*above 2.0 for low AOD stations) from all-point measurements.*

We have also revised the description about the measures to ensure data continuity in Sect.
2.1:

*To ensure adequate records and data continuity in trend analysis, we require the*
*data to have at least 10 years of records with no less than 8 monthly*
*measurements for each year during the 2000-2022 period. Years with less than 8*
*monthly data and seasons with less than 10 years of records are discarded due to*
*poor annual and seasonal representation. We also remove the first or last several*

*months from the time series of certain stations (Canberra and Ilorin), where*
*discontinuities were identified relative to adjacent monthly records.*

**References**

Balarabe, M., Abdullah, K., & Nawawi, M. (2016). Seasonal variations of aerosol optical
properties and identification of different aerosol types based on AERONET data over
sub-sahara west-africa. *Atmospheric and Climate Sciences*, *06*(01), 13–28.
https://doi.org/10.4236/acs.2016.61002

Collaud Coen, M., Andrews, E., Bigi, A., Martucci, G., Romanens, G., Vogt, F. P. A., &
Vuilleumier, L. (2020). Effects of the prewhitening method, the time granularity, and the
time segmentation on the mann–kendall trend detection and the associated sen's slope.
*Atmospheric Measurement Techniques*, *13*(12), 6945–6964. https://doi.org/10.5194/amt-
13-6945-2020

Habib, A., Chen, B., Khalid, B., Tan, S., Che, H., Mahmood, T., et al. (2019). Estimation
and inter-comparison of dust aerosols based on MODIS, MISR and AERONET retrievals
over asian desert regions. *Journal of Environmental Sciences*, *76*, 154–166.
https://doi.org/10.1016/j.jes.2018.04.019

Hirsch, R. M., & Slack, J. R. (1984). A nonparametric trend test for seasonal data with
serial dependence. *Water Resources Research*, *20*(6), 727–732.
https://doi.org/10.1029/wr020i006p00727

Hirsch, R. M., Slack, J. R., & Smith, R. A. (1982). Techniques of trend analysis for
monthly water quality data. *Water Resources Research*, *18*(1), 107–121.
https://doi.org/10.1029/wr018i001p00107

Kulkarni, A., & Storch, H. von. (1992). Simulationsexperimente zur wirkung serieller
korrelation auf den mann-kendall trend test. *Meteorologische Zeitschrift*, *4*(2), 82–85.
https://doi.org/10.1127/metz/4/1992/82

Li, J., Carlson, B. E., Dubovik, O., & Lacis, A. A. (2014). Recent trends in aerosol
optical properties derived from AERONET measurements. *Atmospheric Chemistry and*
*Physics*, *14*(22), 12271–12289. https://doi.org/10.5194/acp-14-12271-2014

Nwofor, O. K., Chidiezie Chineke, T., & Pinker, R. T. (2007). Seasonal characteristics of
spectral aerosol optical properties at a sub-saharan site. *Atmospheric Research*, *85*(1), 38–
51. https://doi.org/10.1016/j.atmosres.2006.11.002

Yue, S., Pilon, P., Phinney, B., & Cavadias, G. (2002). The influence of autocorrelation
on the ability to detect trend in hydrological series. *Hydrological Processes*, *16*(9), 1807–
1829. https://doi.org/10.1002/hyp.1095